# Semi-Supervised Learning with Decision Trees: Graph Laplacian Tree Alternating Optimization

**Arman Zharmagambetov**
Dept. of Computer Science and Engineering
University of California, Merced
`azharmagambetov@ucmerced.edu`

**Miguel Á. Carreira-Perpiñán**
Dept. of Computer Science and Engineering
University of California, Merced
`mcarreira-perpinan@ucmerced.edu`

## Abstract

Semi-supervised learning seeks to learn a machine learning model when only a small amount of the available data is labeled. The most widespread approach uses a graph prior, which encourages similar instances to have similar predictions. This has been very successful with models ranging from kernel machines to neural networks, but has remained inapplicable to decision trees, for which the optimization problem is much harder. We solve this based on a reformulation of the problem which requires iteratively solving two simpler problems: a supervised tree learning problem, which can be solved by the Tree Alternating Optimization algorithm; and a label smoothing problem, which can be solved through a sparse linear system. The algorithm is scalable and highly effective even with very few labeled instances, and makes it possible to learn accurate, interpretable models based on decision trees in such situations.

## 1 Introduction

Semi-supervised learning (SSL) is an important subfield of machine learning which has received a lot of attention in recent years given today's growing amount of data and widespread deployment of machine learning systems. One of the major reasons is that SSL is applicable when labels are scarce. This is in contrast to the traditional fully supervised learning, which requires access to a large amount of high-quality labeled data. However, obtaining such samples is often costly, time-consuming and sometimes even impractical. Therefore, SSL methods have received much praise in the machine learning literature [40] and they are widely used in many applications. A common strategy in SSL is to assume that similar instances have similar predictions, which is commonly incorporated into an objective as a graph prior (e.g. graph Laplacian).

In this paper, we study the problem of training a decision tree model by leveraging a small percentage of labeled data and a much larger sample of unlabeled data. *Why trees?* First, trees are considered to be interpretable models, since the prediction is obtained by routing an input along a unique root-to-leaf path, which can be reformulated as if-then rules. Second, they are widely used in a wide spectrum of applications, such as data mining [30], computer vision [13], finance, etc. They are typically employed as base learners in an ensemble (e.g. bagging [3] or boosting [15]). However, they are used as standalone predictors as well. For instance, recent algorithmic advances in training non-greedy trees have shown that oblique trees can perform competitively well in a number of tasks and strike a good balance between accuracy and interpretability [6, 5, 16–18, 34, 36].

However, like many non-linear methods, decision trees are well known to overfit for small-sized (labeled) data, which is the case in SSL. As an illustration, consider fig. 1, that shows a synthetic binary classification problem in 2D. An oblique tree achieves a certain good performance when it is provided the entire population of labeled data. But the error significantly increases if a tree is trained on six labeled instances only (plot 3). Whereas the benefit is evident when we provide all

36th Conference on Neural Information Processing Systems (NeurIPS 2022).

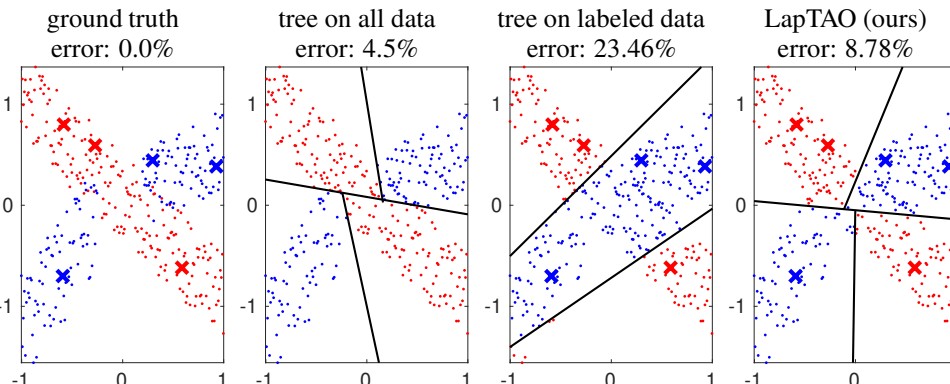

Figure 1: Binary classification on 2D. Plot 1 shows the original data and corresponding class labels. Cross markers ($\times$, six in total) indicate the labeled points that we provide to any given SSL algorithm. "Tree on all data" shows the decision boundary obtained by using all available labeled data, whereas plot 3 uses only six labeled points to train a tree. Plot 4 shows the result of our SSL framework. All trees are oblique of depth $\Delta = 2$.

data (six labeled and the rest are unlabeled) and properly optimize a tree within SSL framework. In our proposed approach, we first state the objective, which consists of a supervised loss (for the labeled data only) and a graph Laplacian regularization (also known as manifold regularization [2]). The resulting optimization problem is long considered to be hard to solve since trees define a non-differentiable, non-convex mapping. By reformulating the problem as a constrained optimization, we derive an efficient and scalable iterative algorithm (section 3) which requires solving two simpler problems at each step: a sparse linear system and a supervised tree learning problem. For the latter, we use the Tree Alternating Optimization algorithm [6, 5], which is crucial for the success of our approach (section 3.1). Moreover, for a special case where the tree structure as well as the parameters in each decision node (not leaves) are fixed, we derive the exact solution given by another linear system (section 3.3). Experimental results (section 4) show the algorithm is able to learn accurate and interpretable decision trees even with very few labeled instances.

## 2 Related work

Although the literature on this topic is immense (see [40, 29]), incorporating decision trees into SSL has received almost no attention. One possible explanation is the difficulty of the optimization problem. Most SSL methods are based on adding a graph prior (or similarity matrix) as regularization to exploit the geometry of the underlying data distribution [2, 39, 41, 42]. Minimizing an objective with such regularization is non-trivial when we have decision trees as a predictive model.

That said, several attempts have been made to apply SSL for trees. Levatić et al. [23] modify a splitting criterion for recursive tree induction by taking into account unlabeled data. That is, a splitting score for each [feature, threshold] pair consists of two parts: the traditional purity score (e.g. Gini index) for labeled data and a "clustering" score for all available data. Tanha et al. [27] apply a self-training framework [33] to train classification trees with labeled and unlabeled data. Note that self-training is a generic framework that can be applied with any classifier which predicts class probabilities. It trains a classifier iteratively where the first iteration includes only labeled data. After that, it uses model predictions and adds the most confidently predicted instances to enrich the amount of supervision. Another direct approach is to apply a label smoothing technique [41] to propagate label information to all data and then fit a tree. This method has been recently reused within the graph neural networks framework to train boosted trees [12, 21]. Finally, Kemp et al. [22] proposed a Bayesian approach where unlabeled data assist in inferring a latent tree structure from a distribution over trees. The limitations of this approach are that it covers classification tasks only, and it uses Markov Chain Monte Carlo to sample from the distribution over trees, which leads to scalability issues.

Unlike all these methods, we attempt to directly minimize a regularized SSL objective (with a graph prior) over the parameters of a decision tree. The method is applicable to both regression and

classification problems, and can train trees of arbitrary type (i.e., axis-aligned, oblique, etc.). In fact, our optimization framework is generally applicable to any type of predictive functions, not just trees.

## 3 LapTAO: semi-supervised learning framework for decision trees

We are given the dataset $\mathcal{D} = \mathcal{D}_l \cup \mathcal{D}_u$, where $\mathcal{D}_l = \{\mathbf{x}_n, y_n\}_{n=1}^l \subset \mathbb{R}^D \times \mathbb{R}$ is the labeled portion of the data, with $l$ points, and $\mathcal{D}_u = \{\mathbf{x}_n\}_{n=l+1}^N \subset \mathbb{R}^D$ is the unlabeled portion, with $N - l$ points. Then our goal is to minimize the following regularized objective:

$$E(\boldsymbol{\Theta}) = \sum_{n=1}^l (T(\mathbf{x}_n; \boldsymbol{\Theta}) - y_n)^2 + \alpha\,\phi(\boldsymbol{\Theta}) + \gamma \sum_{n,m=1}^N w_{nm}(T(\mathbf{x}_n; \boldsymbol{\Theta}) - T(\mathbf{x}_m; \boldsymbol{\Theta}))^2. \quad (1)$$

Here, $w_{nm}$ are the weights in the affinity (similarity) matrix based on a graph on all the data points $\mathcal{D}$, usually a nearest-neighbor graph; $T \colon \mathbb{R}^D \to \mathbb{R}$ is the tree predictive mapping, with parameters $\boldsymbol{\Theta} = \{\boldsymbol{\theta}_i\}_{\text{nodes}}$; $\phi(\cdot)$ is a regularization penalty, such as $\|\cdot\|_1$; and $\gamma, \alpha$ are regularization hyperparameters. Rather than using a greedy recursive partitioning procedure (such as CART [4] or C5.0 [26]), which does not optimize any loss function, we consider $T$ as a parametric model with trainable weights in each node (like fixing a neural net architecture and optimizing over its parameters). If $T$ was differentiable, one could optimize (1) via gradient-based methods, as can be done for neural nets [31]. Similarly, the solution is relatively straightforward to obtain if problem (1) is convex [2]. However, solving problem (1) is non-trivial with a tree which defines a non-differentiable and non-convex mapping. Instead, we apply the *method of auxiliary coordinates* [8, 9], a generic method for optimizing nested systems.

We proceed as follows by reformulating problem (1) in an equivalent form (similar to that in [7, 36, 17] for dimensionality reduction and clustering). Introduce a new auxiliary variable $z_n \in \mathbb{R}$ for each training instance $n$ and consider the constrained problem:

$$\min_{z_1,\ldots,z_N,\boldsymbol{\Theta}} \sum_{n=1}^l (z_n - y_n)^2 + \alpha\,\phi(\boldsymbol{\Theta}) + \gamma \sum_{n,m=1}^N w_{nm}(z_n - z_m)^2 \quad (2)$$

$$\text{s.t.} \quad z_n = T(\mathbf{x}_n; \boldsymbol{\Theta}) \quad n = 1,\ldots,N. \quad (3)$$

Obviously, by putting the constraints (3) back into eq. (2), we end up with the same objective as in (1), so these two problems are equivalent. Let us denote $\mathbf{y} = [y_1, y_2, \ldots, y_l, 0, 0, \ldots]^T \in \mathbb{R}^N$ the augmented ground truth vector, i.e., we put zeros in the unlabeled portion of the data. Similarly, introduce a diagonal matrix $\mathbf{J} = \mathrm{diag}(1,\ldots,1,0,\ldots,0) \in \mathbb{R}^{N \times N}$ with the first $l$ diagonal entries equal to 1 and the rest to 0. Also, let the graph Laplacian be $\mathbf{L} = \mathbf{D} - \mathbf{W}$ with a diagonal matrix $\mathbf{D} \in \mathbb{R}^{N \times N}$ (the degree matrix) having entries $d_{nn} = \sum_{m=1}^N w_{nm}$, and let $\mathbf{W} = (w_{nm}) \in \mathbb{R}^{N \times N}$ be the affinity matrix. Finally, call $\mathbf{z} = [z_1,\ldots,z_N]^T$ and $\mathbf{t}(\mathbf{X}; \boldsymbol{\Theta}) = [T(\mathbf{x}_1; \boldsymbol{\Theta}),\ldots,T(\mathbf{x}_N; \boldsymbol{\Theta})]^T$, where $\mathbf{X} = (\mathbf{x}_1,\ldots,\mathbf{x}_N)$. Then we can rewrite eq. (2)-(3) as follows:

$$\min_{\mathbf{z},\boldsymbol{\Theta}} (\mathbf{z} - \mathbf{y})^T \mathbf{J}\,(\mathbf{z} - \mathbf{y}) + \alpha\,\phi(\boldsymbol{\Theta}) + \gamma\,\mathbf{z}^T \mathbf{L}\,\mathbf{z} \quad \text{s.t.} \quad \mathbf{z} = \mathbf{t}(\mathbf{X}; \boldsymbol{\Theta}). \quad (4)$$

Now we solve this using the augmented Lagrangian method [25]. This defines a new, unconstrained optimization problem:

$$\min_{\mathbf{z},\boldsymbol{\Theta}} (\mathbf{z} - \mathbf{y})^T \mathbf{J}\,(\mathbf{z} - \mathbf{y}) + \alpha\,\phi(\boldsymbol{\Theta}) + \gamma\,\mathbf{z}^T \mathbf{L}\,\mathbf{z} - \boldsymbol{\lambda}^T(\mathbf{z} - \mathbf{t}(\mathbf{X}; \boldsymbol{\Theta})) + \mu\|\mathbf{z} - \mathbf{t}(\mathbf{X}; \boldsymbol{\Theta})\|^2 \quad (5)$$

where $\boldsymbol{\lambda} \in \mathbb{R}^N$ are the estimates of the Lagrange multipliers. Optimizing this for each $\mu > 0$ produces a sequence of $(\mathbf{z}_\mu, \mathbf{t}_\mu(\mathbf{X}; \boldsymbol{\Theta}))$ and, as $\mu \to \infty$, we gradually force the minimizer to be in the feasible set of the constrained problem. Finally, in order to minimize (5) over $\mathbf{z}$ and $\mathbf{t}(\mathbf{X}; \boldsymbol{\Theta})$ for fixed $\mu$, we apply alternating optimization over $\mathbf{z}$ and $\boldsymbol{\Theta}$:

- **Label-step** (optimizing over $\mathbf{z}$ given fixed $\mathbf{t}(\mathbf{X}; \boldsymbol{\Theta})$). The objective in eq. (5) is a quadratic function and a minimizer is obtained by solving the linear system:

$$\min_{\mathbf{z}} (\mathbf{z} - \mathbf{y})^T \mathbf{J}\,(\mathbf{z} - \mathbf{y}) + \gamma\,\mathbf{z}^T \mathbf{L}\,\mathbf{z} - \boldsymbol{\lambda}^T(\mathbf{z} - \mathbf{t}(\mathbf{X}; \boldsymbol{\Theta})) + \mu\|\mathbf{z} - \mathbf{t}(\mathbf{X}; \boldsymbol{\Theta})\|^2 \Rightarrow$$

$$\mathbf{A}\mathbf{z} = \mathbf{J}\mathbf{y} + \mu\mathbf{t}(\mathbf{X}; \boldsymbol{\Theta}) + \frac{1}{2}\boldsymbol{\lambda} \quad (6)$$

where $\mathbf{A} = \mathbf{J} + \mu\mathbf{I} + \gamma\mathbf{L}$ is a positive definite matrix. Moreover, $\mathbf{A}$ is a sparse matrix if the graph Laplacian $\mathbf{L}$ is sparse, which is the case in practice if we construct $\mathbf{W}$ by using a nearest neighbors graph. This allows us to solve a large scale linear system in an efficient way (e.g. by caching a matrix factorization or using the conjugate gradient method). Intuitively, the label-step can be interpreted as approximating the labels (for $\mathcal{D}_u$) using the graph Laplacian and predictions obtained from the current tree (i.e., label smoothing).

- **Tree-step** (optimizing over $\boldsymbol{\Theta}$ given fixed $\mathbf{z}$). Problem (5) reduces to a regression fit of a tree:

$$\min_{\boldsymbol{\Theta}} \; \mu\|\mathbf{z} - \mathbf{t}(\mathbf{X};\boldsymbol{\Theta})\|^2 + \alpha\,\phi(\boldsymbol{\Theta}) - \boldsymbol{\lambda}^T(\mathbf{z} - \mathbf{t}(\mathbf{X};\boldsymbol{\Theta})) \Leftrightarrow$$

$$\min_{\boldsymbol{\Theta}} \; \left\|\left(\mathbf{z} - \frac{1}{2\mu}\boldsymbol{\lambda}\right) - \mathbf{t}(\mathbf{X};\boldsymbol{\Theta})\right\|^2 + \frac{\alpha}{\mu}\phi(\boldsymbol{\Theta}). \tag{7}$$

  Note that here we use $(\mathbf{z} - \frac{1}{2\mu}\boldsymbol{\lambda})$ as "ground-truth" labels (not $y_n$, which is not defined for $\mathcal{D}_u$ anyway). We solve this problem using the Tree Alternating Optimization (TAO) algorithm, which we describe in section 3.1 below. Intuitively, this step can be understood as fitting a tree with the current estimates of the labels.

Finally, the step over Lagrange multipliers is done by the update $\boldsymbol{\lambda} \leftarrow \boldsymbol{\lambda} - \mu(\mathbf{z} - \mathbf{t}(\mathbf{X};\boldsymbol{\Theta}))$. In summary, our algorithm alternates between solving a linear system and training a tree. After each (label,tree)-step, we increase the penalty parameter $\mu$, we update $\boldsymbol{\lambda}$ and we keep iterating until approximate convergence or other stopping criterion (e.g. maximum number of iterations reached). We call our algorithm *LapTAO* and provide detailed pseudocode in the suppl. mat.

**Initialization for LapTAO**   To start our iterative algorithm, we need to obtain initial solutions $(\mathbf{z}_0, \mathbf{t}_0)$ for eq. (5) when $\mu \to 0^+$. This is straightforward to achieve for $\mathbf{z}_0$, as it involves solving the same linear system as in eq. (6) but with $\mu = 0$ and $\boldsymbol{\lambda} = \mathbf{0}$: $(\mathbf{J} + \gamma\mathbf{L})\mathbf{z} = \mathbf{J}\mathbf{y}$. This can be considered as a non-parametric smoothing of the labels obtained by propagating (diffusing) the ground-truth labels over all points (labeled and unlabeled) through the graph Laplacian. Although these smoothed labels are not optimal in problem (1), which requires optimizing them jointly with the tree, they do provide a good initialization, and it is convenient to solve the linear system exactly for $\mu = 0$. After that, we fit a tree using $\mathbf{z}_0$ as ground-truth labels (tree-step). Note that TAO requires an initial tree (see section 3.1), which we obtain by generating a complete tree of depth $\Delta$ (a hyperparameter) with a Gaussian random weight vector at each decision node.

**Hyperparameters of LapTAO**   The hyperparameters are $\gamma$ for the graph prior and $\alpha$ (sparsity) and $\Delta$ (depth) for the tree. They can be selected by cross-validation.

**Extension to multioutput regression and classification**   We can extend LapTAO for multiple outputs in a straightforward way: the label-step solves the linear system (6) for each dimension separately and the tree-step applies TAO as usual, as it can handle a vector-valued output. For classification, we use a one-hot encoding of the labels and consider the problem as a regression task, as is commonly done in the decision tree literature [19] (especially for boosted trees). It is possible to extend our framework for other losses (e.g. hinge, logistic) that may work better for classification. This requires certain changes in the label-step and we will explore it in our future works.

**Extension to models other than trees**   Although our focus in this paper are decision trees, the semi-supervised learning optimization algorithm we propose is perfectly general. The model $(T(\mathbf{x};\boldsymbol{\Theta}))$ appears in the algorithm in the tree-step, with the form of a regression problem having the smoothed labels as ground-truth. Obviously, we can use other regression models, such as random forests, gradient boosted trees, neural networks, etc. The motivation to use our approach was the fact that trees are not differentiable, so one cannot optimize problem (1) by gradient-based methods. But our approach has another, computational advantage: by separating terms through the auxiliary variables $\mathbf{z}$, the quadratic-cost term is confined to the label-step. For large datasets, this is a sparse linear system, for which efficient algorithms exist. Hence, the complexity associated with the model (tree) is linear on the dataset size (in the tree-step). This is much faster than having to deal directly with the quadratic term *and* the model, as in eq. (1), whether for trees or other models.

## 3.1   Overview of the tree alternating optimization (TAO) algorithm

Potentially, one could apply any tree fitting algorithm to solve the tree-step in LapTAO, such as CART [4], C5.0 [26], OC1 [24], etc. (we do show such results in the suppl. mat.). But there

are several important considerations. Firstly, from an optimization point of view, it is known that alternating optimization is most effective when the step over each block is (ideally) exact. This is computationally achievable for the label-step, which involves a linear system. However, training a tree optimally even in the simplest case (axes-aligned with binary inputs and output) is NP-hard [20]. Therefore, we need an approximate but good solution. Most traditional tree learning algorithms, based on greedy recursive partitioning (such as CART), are highly suboptimal [19] and do not even consider any specific loss function over trees. In contrast, the Tree Alternating Optimization (TAO) algorithm [6, 5] fits decision trees by monotonically decreasing a well-defined and very general loss function and regularization over a well-defined parametric space of trees, given an initial tree structure and parameter values. This makes it also possible to use warm-start in the tree-step, i.e., to continue improving the tree from the previous iteration—which greedy recursive partitioning cannot do, as it constructs a new tree from scratch every time. Warm-start is essential to speed up the optimization and to achieve stability in the results (CART-type algorithms are notoriously unstable in that small changes in the training set can result in drastically different tree structures and parameters [19]). A further, important advantage of TAO is that it can learn trees of quite general types, such as oblique trees (see section 3.2), which are far more powerful that the traditional axes-aligned trees.

TAO has been shown to find much better trees under a variety of losses, regularization and types of tree, as well as forests, and scales well to large datasets, see e.g. [37, 34, 38, 35, 16, 10]. Below we provide a summary of TAO for completeness; more details can be found in [6, 5, 34]. TAO considers a decision tree as a parametric model with a given structure and trainable parameters in each node ($\boldsymbol{\Theta} = \{\boldsymbol{\theta}_i\}_{\text{nodes}}$), and it optimizes an objective consisting of a loss and a regularization term, such as that in eq. (7). TAO requires the loss to be a sum over individual points, so it cannot handle eq. (1), which includes pairwise distances. However, our reformulation of the problem allows us to apply TAO since the minimization in eq. (7) is now in the desirable form.

TAO takes as input an initial tree, for example a complete tree of depth $\Delta$ with random weights at each node, or a tree created by another algorithm. It uses a *separability condition* to decompose the loss function (7) over subsets of non-descendant nodes. For instance, all nodes at the same depth are non-descendant w.r.t. each other and can be trained independently (and in parallel). Then, TAO uses an alternating optimization scheme such as the following: 1) pick all nodes at a certain depth and fix all remaining nodes (i.e., we consider them as a function with constant parameters); 2) optimize each node at that depth efficiently by solving a *reduced problem* (see below); 3) repeat this for all depths. One pass through all nodes is a TAO iteration, and we keep iterating until convergence occurs or until we reach a maximum number of iterations (see pseudocode in the suppl. mat.). Optimizing a single node $i$ (decision node or leaf) over its parameters can be shown to be equivalent to a simpler, reduced problem operating only on those training points which currently reach $i$ (denoted as *reduced set $\mathcal{R}_i$*). The solution of the reduced problem depends on the type of node:

- **Decision node**. With binary trees, this involves solving a certain binary classification problem. Given the reduced set $\mathcal{R}_i$, we pass each input $\{\mathbf{x}_n, y_n\} \in \mathcal{R}_i$ to both children of $i$ and calculate the error ($e_n^{\text{left}}, e_n^{\text{right}}$) induced by each child (note that all descendant nodes and their parameters are fixed). Next, assign a pseudolabel $\bar{y}_n \in \{-1, 1\}$ depending on which child brings the lower error (either left or right). Each pseudolabel comes with the weight $|e_n^{\text{left}} - e_n^{\text{right}}|$ since the error made for each instance is different. Solve a weighted 0/1 loss binary classification on $\{\mathbf{x}_n, \bar{y}_n\}$.

- **Leaf**. The actual prediction of a tree occurs in its leaves; the decision nodes are just responsible for routing a point to the corresponding leaf. Therefore, the optimization problem for a leaf is equivalent to the original problem (7) but on its reduced set. Its solution depends on what type of leaf model we use. With a constant label, it is the average of the response values ($y_n$) in that leaf. With linear leaves, the solution is to fit a linear regressor on $\mathcal{R}_i$ (possibly with regularization).

## 3.2 Sparse oblique decision trees

The TAO algorithm is applicable to a large spectrum of decision tree types. Here we pick a *sparse oblique tree with constant leaves* as our main model. Each decision node $i$ makes a hyperplane-based split: go to the left child if $\mathbf{w}_i^T \mathbf{x} < w_{i0}$, else go to the right one. Therefore, the decision node reduced problem (which is NP-hard) is approximated by fitting a logistic regression. Additionally, we apply an $\ell_1$ penalty as regularization term $\phi(\cdot)$ to encourage sparsity. We use LIBLINEAR [14] to fit this model. As for the leaves, we take the average of the response values ($y$) in that leaf.

The motivation behind choosing sparse oblique trees is twofold. First, traditional axis-aligned trees are very restrictive since each split uses a single feature, which neglects interactions or correlations between features. Indeed, empirical results show oblique trees achieve far better performance [38]. Besides, thanks to the sparsity, only few features are active at each split, and nodes in the initial tree can become redundant and be pruned if their weight $\mathbf{w}_i$ becomes zero [6]. This, together with the fact that oblique trees typically have small depth, makes the final model more interpretable.

### 3.3 Special case: exact solution when the tree structure is fixed

Problem (1) is hard to solve over the entire space of decision trees, but we can obtain an exact solution if the structure and decision node parameters are fixed, since then the problem reduces to solving a linear system. In this case, the only parameters to optimize are in the leaves. Assuming each leaf outputs a constant value, we can reformulate the tree prediction as a sum of basis functions $T(\mathbf{x}) = \sum_{i=1}^{m} c_i \, b_i(\mathbf{x})$, where $m$ is the number of leaves, $c_i \in \mathbb{R}$ is leaf $i$'s label and $b_i(\cdot) \in \{0, 1\}$ is 1 only if $\mathbf{x}$ ends up in leaf $i$. Now we can rewrite (1) as the following minimization problem over the parameters of all the leaves $\mathbf{c} = (c_1, \ldots, c_m)^T$:

$$E(\mathbf{c}) = \sum_{n=1}^{l} \left( \sum_{i=1}^{m} c_i \, b_i(\mathbf{x}_n) - y_n \right)^2 + \gamma \sum_{n,m=1}^{N} w_{nm} \left( \sum_{i=1}^{m} c_i \, (b_i(\mathbf{x}_n) - b_i(\mathbf{x}_m)) \right)^2 \quad (8)$$

$$= (\mathbf{Bc} - \mathbf{y})^T \mathbf{J} \, (\mathbf{Bc} - \mathbf{y}) + \gamma \, \mathbf{c}^T \mathbf{B}^T \mathbf{LBc} \quad (9)$$

where $\mathbf{B} = (b_i(\mathbf{x}_n)) \in \mathcal{R}^{N \times m}$ can be precomputed since we fix the tree structure and parameters in all decision nodes. Minimizing this over $\mathbf{c}$ yields the following linear system:

$$\mathbf{Ac} = \mathbf{B}^T \mathbf{Jy} \quad (10)$$

where $\mathbf{A} = \mathbf{B}^T \mathbf{JB} + \gamma \, \mathbf{B}^T \mathbf{LB}$ is a matrix of $m \times m$. This is very fast to solve since oblique trees are quite shallow, so the number of leaves $m$ is not large (at most 1 000 in our experiments). Once LapTAO is finished, we apply the above procedure as a post-processing to the final tree.

### 3.4 Computational complexity of LapTAO

At the top level, LapTAO runs a fixed number of iterations (depending on the $\mu$ schedule, typically less than 20). Each iteration has to solve (approximately) two subproblems:

- **Label-step**: this is a large, sparse linear system of $N \times N$ (where $N$ is the sample size). We solve it approximately with conjugate gradients (CG), initialized by the previous iterate (warm-start). Each CG iteration is $O(Nk)$ where $k$ is the average number of neighbors in the graph, and we run just a few CG iterations. The total runtime of the label step is less than 30 seconds in the largest experiment we conducted (1M points). Convergence can be further improved via preconditioning (e.g. Jacobi). We can also solve the linear system exactly in $O(N^2)$ by caching its SVD, as noted in the suppl. mat. (section 3), but this is only convenient if $N$ is a few thousands at most.

- **Tree-step**: fitting an oblique tree with TAO to the $N$ training points. Each iteration of TAO updates each decision node and leaf node. For each leaf, we compute the average of the labels of its reduced set (training points reaching it), so this is $O(N)$ over all the leaves. For each decision node, we train a logistic regression on its reduced set. Assuming logistic regression is linear on the sample size and dimensionality, this is $O(ND)$ total for all the decision nodes at the same depth, although with a larger constant factor in the big-O notation than for the leaves. Hence, processing all the decision nodes in the tree is $O(\Delta ND)$, or equivalently, running $\Delta$ logistic regressions on the whole training set. See more details in [6, 5]. A critical computational advantage of TAO is due to the fact that each node (decision or leaf node) only handles the points in its reduced set. Therefore, TAO itself can be parallelized depthwise. In summary, the overall runtime of TAO is $O(\Delta ND)$ per TAO iteration. We run 10 TAO iterations in our experiments.

Since the tree-step dominates the label-step, in terms of runtime our algorithm is almost like sequentially training decision trees (as in boosting). Additionally, each tree-step can be parallelized. Further acceleration can be done using GPUs. This is possible with GPU-friendly implementations of logistic regression, and also because oblique trees involve scalar products (unlike axis-aligned trees).

Finally, this computational cost should also include computing the nearest-neighbor graph and its affinity matrix $\mathbf{W}$. This is indeed a large cost, and it affects all semi-supervised learning methods based on the graph Laplacian. A naive implementation requires $O(DN^2)$ to calculate the distance vector for each point and determine the nearest neighbors. For large datasets, one usually uses approximate nearest neighbors (e.g. via Locality Sensitive Hashing or other techniques [1]).

In terms of storage, besides the tree $T$, we need to store the auxiliary variables $\mathbf{z}$ (predicted labels) for the entire training set, which are $N$ scalars for 1D regression or binary classification.

## 4   Experiments

This section shows our experimental findings. We demonstrate that *the proposed method dominates over other semi-supervised learning frameworks in accuracy and approaches fully supervised baseline with far less amount of labeled data*. This is true with practically no exceptions against baseline tree-based models where accuracy margin is often quite large. As for the other methods (non tree-based), we either outperform them or achieve similar error, which makes LapTAO a strong competitor. To show that, we consider several regression and classification benchmarks of varying sizes and across different domains. We were able to run our algorithm on a dataset with up to 1 million instances on a regular PC, which shows its scalability. Moreover, using fashion-mnist as an example, we demonstrate that the final model, a shallow oblique tree with sparse parameter vector in each node, provides insights into how it achieves a prediction allowing model interpretability.

### 4.1   Experimental setup

We compare our proposed approach (LapTAO) with the following baselines: 1) *oblique–all* fits an oblique tree with full supervision, this shows the theoretical maximum performance we can achieve; 2) *oblique–lbl* is the oblique trees trained on labeled portion of data $\mathcal{D}_l$ (this completely discards large portion of unlabeled data); 3) Self-training (*axis–self*, *oblique–self*) is an iterative procedure that uses the model predictions to enlarge the portion of labeled data (see section 2 for details and references), we closely follow the implementation by Yarowsky [33]; here, "axis" means traditional axis-aligned trees; 4) Laplacian SVM is a seminal work by Belkin et al. [2] which has similar problem formulation as in eq. (1) but for SVM. Additionally, the suppl. mat. include the comparison with semi-supervised classification trees by Levatić et al. [23] and EBBS by Chen et al. [12]. Regarding hyperparameters, given the fixed cross-validation set (1% of train data), we explored as best as we could all important hyperparameters for all methods (see details in the suppl. mat.). These include: controlling a tree depth ($\Delta$), confidence threshold for self-training, $\sigma$ and $C$ values for LapSVM, etc. It worth to mention that the hyperparameter settings suggested by authors or their default values work best in most cases.

We use TAO to train oblique trees and CART [4] to train axis-aligned trees. For all methods that use TAO, we set the total number of TAO iterations to 15. The depth $\Delta$ as well as the regularization parameter $\alpha$ are tuned via cross-validation. As for the settings that are specific to LapTAO, we proceed as follows. To construct the graph Laplacian, we use the Gaussian affinities with $k$-nearest neighbors and perplexity parameter $\mathcal{K}$ tuned for each dataset. The linear system in the label-step is solved either using direct methods (less than 20k dimensions) or Conjugate Gradient method for large scale problems. We use $\gamma = 0.1$ in all experiments. As for the main loop of the augmented Lagrangian, we iterate 20 times starting from small value for $\mu_0 = 0.001$ multiplied by 1.5 after each iteration. The remaining details as well as dataset descriptions can be found in the suppl. mat.

### 4.2   Main results

Fig. 2 summarizes the main results which are the trade-off plots of test error versus the percentage of labeled data on two regression and classification tasks. Intuitively, the error should go down monotonically as we increase the amount of supervision which is clearly the case in all figures. According to our findings, KNN and "axis–self" show the worst results in almost all benchmarks. The only case when KNN performed reasonably good was on MNIST, which is known to work well with "template classifiers" (e.g. RBF network, kernel SVM, KNN, etc.). Even in that case it has a large error gap with respect to LapTAO. The poor performance of the "axis–self" can be explained by suboptimality of greedily grown trees [19] and suboptimality of the self-training approach, which is mostly based on heuristics. Next, oblique trees trained on supervised data only ("oblique–lbl")

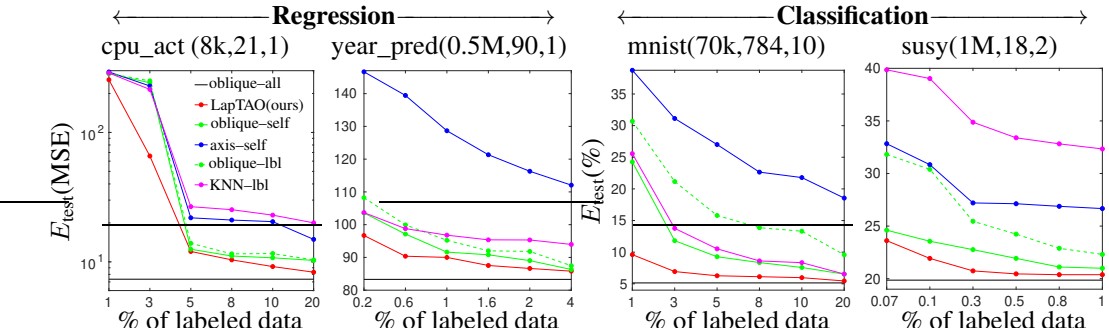

Figure 2: Results on regression (year_pred, cpu_act) and classification(mnist, susy) tasks. Numbers in brackets report the training size, number of features and output dimension (or number of classes). x–axis shows the percentage of labeled data provided to the algorithm and y–axis shows the test error. Baselines: *oblique–all* fully supervised baseline (i.e., trains an oblique tree on 100% of labeled data); *\*–lbl* uses $\mathcal{D}_l$ only to train the corresponding model; *\*–self* is an iterative self-training approach (see section 2 and 4.1).

leads to the significant drop in accuracy (magenta vs black lines). This shows that relying only on labeled data is not enough to achieve a decent performance. Incorporating an oblique tree into self-training framework brings certain benefits ("oblique–self"), notably for classification tasks (green dashed vs solid lines). Finally, *LapTAO consistently improves over all other SSL methods*, often by a considerable margin. For instance, in case of 3% in cpu_act and 1% in MNIST, the difference in the error with the second best SSL approach is several orders of magnitude. It shows acceptable results even in extreme label scarcity scenarios, e.g. when we provide $< 0.5\%$ of labeled data on year_pred and susy. Moreover, LapTAO approaches the fully supervised baseline more quickly: for MNIST, we can achieve the same $\sim 5\%$ test error as "oblique–all" using only 20% of labeled training points.

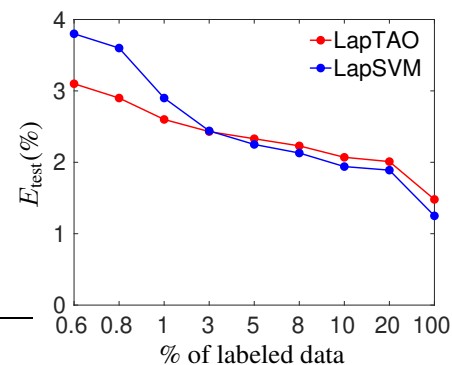

Figure 3: Comparison against LapSVM on Fashion-MNIST (3 classes: "shirt", "bag" and "ankle boot").

**Comparison with Laplacian SVM** LapSVM is a natural baseline to compare with since it uses the same problem formulation as in eq. (1) but for the support vector machines. However, we were not able to include it to the previous comparison (in fig. 2) due to implementation issues: 1) it is impractical to apply it for problems beyond 30k instances as it requires computing the inverse of the modified dense Gram matrix on the entire dataset;[1] 2) it can handle a classification task only. Therefore, we pick the subset of Fashion-MNIST (3 classes: "shirt", "bag" and "ankle boot") resulting in 18k training points. To make the comparison as fair as possible, both of the algorithms use the same graph Laplacian matrix and we enforce the same penalty on it ($\gamma = 0.1$). For LapSVM, we use the rbf kernel with $\sigma = 5$ and the hyperparameter for LIBSVM [11] is set to $C = 100$.

The results are illustrated in fig. 3. It is worth to mention that similar to the original MNIST, "template classifiers", such as kernel SVM, show quite good performance on this task which makes LapSVM a strong baseline [32]. On top of that, the problem formulation in eq. (1) is still convex for SVM and can be efficiently solved, whereas we are dealing with much harder problem for oblique trees. Therefore, it is nice to see that LapTAO performs similarly (but slightly worse) up to some point. However, it is surprising to see that the error gap between our approach and LapSVM narrows as we introduce more label scarcity and eventually we start to outperform when % of labeled data = 3%. From that point on, the table turns to the side of LapTAO and the difference becomes more and more noticeable (especially for 0.6%). One possible explanation for this behavior is the overfitting issue on small datasets, which

---

[1]One could approximate the inverse by Nyström (or other) method but this is beyond the scope of this paper.

Table 1: Training runtime for different semi-supervised learning algorithms (in seconds).

| Dataset \ Method | LapTAO | oblique–self | axis–self | SSCT |
|---|---|---|---|---|
| cput_act | 1072s | 934s | 23s | 936s |
| mnist | 11027s | 9572s | 514s | 15932s |
| susy | 24578s | 17873s | 816s | >1d |

is a known problem for kernel SVM, whereas sparse oblique trees are shown to be relatively robust to that [38].

**Training time**    Table 1 reports the training time for different baseline methods. Overall, LapTAO algorithm for the largest experiment we performed (on susy) took less than 7 hours and around 3 hours for the moderate dataset size (mnist). This is comparable to the self-training baseline (i.e., oblique–self) but LapTAO produces far better trees in terms of accuracy. Please note that we ran our code on a regular PC (Intel(R) Core(TM) i7-7700 CPU @ 3.60GHz, 32GB RAM), with little parallel processing and using unoptimized Python implementation. Therefore, the training runtime for LapTAO can be significantly improved. We did not use any GPUs.

### 4.3   Model interpretability

Model explainability and interpretability is a topic of renewed interest due to the widespread usage of machine learning and the risks associated with privacy, algorithmic bias, etc. In order to trust and rely on such automated systems, it is crucial to understand how they achieve a certain prediction. In contrast to "black box" models, decision trees are long considered as interpretable models due to the hierarchical structure. This allows to transform the model prediction as "if-then" rules extracted from root-to-leaf path. Specifically for oblique trees, each logical clause takes the following form: go to left child if $\mathbf{w}^T\mathbf{x} < w_0$, else go to right. This makes the interpretation a little harder since we need to look at linear combination of features at each split. However, in our case, we add $\ell_1$ penalty which encourages parameter vector at each node to be sparse, i.e., only few features participate in decision making.

In this section, we argue that the oblique trees trained using LapTAO strike a good trade-off between accuracy and interpretability which is controlled via hyperparameter $\alpha$. To illustrate this, we use the same subset of Fashion-MNIST as in section 4.2 and train a sparse oblique tree using LapTAO (10% of training data are labeled). By decreasing the value of $\alpha$ we enforce more sparsity resulting into shallower and more interpretable trees (since complexity decreases). However, we sacrifice the performance since the error goes up.

Fig. 4 shows the results for $\alpha = 1$ and $\alpha = 10$, more results can be found in the suppl. mat. For simplicity, let us focus on the bottom tree ($\alpha = 10$). Clearly, each leaf contains instances of nearly the same class since the average image looks like a representative "template" from the corresponding class. As for the decision node, consider the root (node #1). All "Boot" images are sent to the right child of the root. Also, it is easy to notice that such images do not contain any pixels in the top-left quadrant of the image. Therefore, the weight vector at the root has negative (blue) values in the corresponding elements and we know that negative values are responsible for sending an instance to the left. In other words, all images that have something in the top-left quadrant are sent to the left. Therefore, boots will end up in the right child. Similarly, node #2 sends most of the images that have M-shaped stroke in the top-center part (e.g. large bags, shirts with collar) to the left child. Following the same logic, we can obtain meaningful insights for each decision node. Also note that all nodes have majority of values equal to zero (thanks to the sparsity) which makes the interpretation easier.

## 5   Conclusion

Semi-supervised learning is most commonly formulated with a graph-based regularization term, which encourages the labels of nearby (similar) points to be similar. This has been very successful with models such as neural nets and kernel machines, but until now it remained an open problem with decision trees, which define a nondifferentiable function. We have shown how to reformulate the problem in a way that is amenable to iterative optimization. By introducing auxiliary variables, we isolate the difficult part (the tree optimization), and all the algorithm needs to do is to fit a regression tree in alternation with solving a sparse linear system that smoothes the predicted labels. The tree

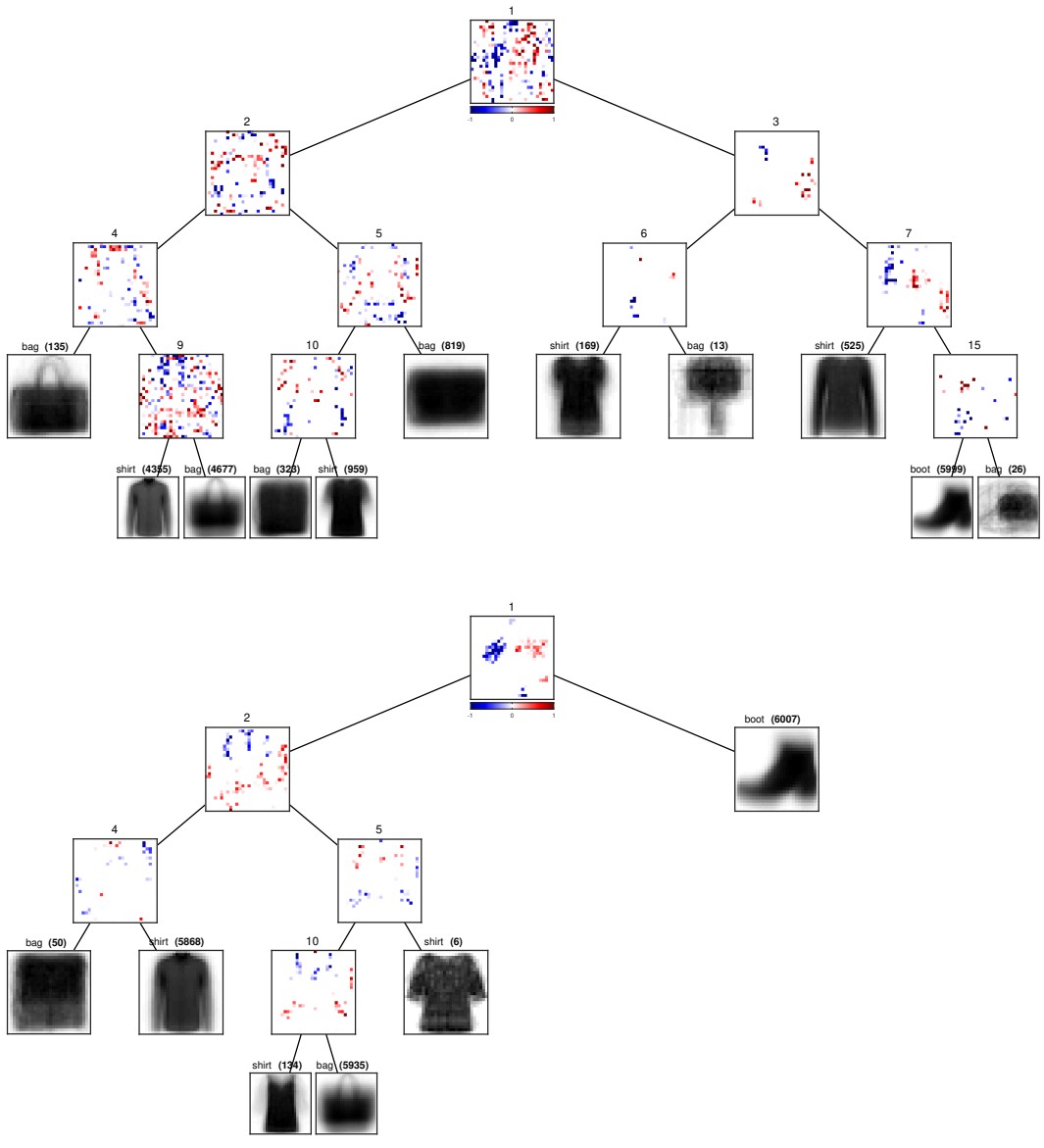

Figure 4: Some of the oblique trees obtained from LapTAO on Fashion-MNIST. Both figures use 10% of labeled data, but they differ in regularization penalty ($\alpha$) on the tree: (*top*) $\alpha = 1$ with $E_{\text{test}} = 2.1\%$ and (*bottom*) $\alpha = 10$ with $E_{\text{test}} = 3.9\%$. At each decision node, we illustrate the weight vector of dimension 784 reshaped into $28 \times 28$ square where each value is colored according to their sign and magnitude (positive, negative and zero values are blue, red, and white, respectively). At each leaf, we show the class label, the total number of training points in that leaf (in brackets), and the average of input images in that leaf (as a greyscale image).

fitting can be done reliably and efficiently using the Tree Alternating Optimizing (TAO) algorithm, which also allows us to use more powerful trees, such as sparse oblique trees. Our experimental results demonstrate that the algorithm can train accurate and interpretable decision trees even in extreme label scarcity situations. Our framework can be generalized to other machine learning models, such as ensembles of trees, which will be the future direction of our work.

## Acknowledgments and Disclosure of Funding

Work partially supported by NSF awards IIS–1423515 and IIS–2007147.

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
