# Supplementary materials for:
# Semi-Supervised Learning with Decision Trees: Graph Laplacian Tree Alternating Optimization

**Arman Zharmagambetov**
Dept. of Computer Science and Engineering
University of California, Merced
azharmagambetov@ucmerced.edu

**Miguel Á. Carreira-Perpiñán**
Dept. of Computer Science and Engineering
University of California, Merced
mcarreira-perpinan@ucmerced.edu

## Abstract

In this supplementary material, we provide the following: 1) Pseudocodes for LapTAO and TAO (section 1); 2) Derivation of the solution for the label-step (section 2); 3) How to accelerate the "label–step" in LapTAO (section 3); 4) Description of the experimental setup: datasets, comparison methods, hyperparameters, etc. (section 4); 5) Additional experimental results (section 5): comparison with SSCT [6] and EBBS, decision tree visualizations, etc.

## 1 Pseudocodes

---

**input** labeled set $\mathcal{D}_l = \{\mathbf{x}_n, y_n\}_{n=1}^l$ and unlabeled set $\mathcal{D}_u = \{\mathbf{x}_n\}_{n=l+1}^N$;
    penalty parameters: $\alpha, \gamma$; $\mu$ schedule: $\mu_0, \ldots, \mu_{\max}$;
    graph Laplacian $\mathbf{L} = \mathbf{D} - \mathbf{W}$;
$\boldsymbol{\lambda} \leftarrow \mathbf{0}$ (initialize Lagrange multipliers);
$\mathbf{z}_0 \leftarrow$ solve the linear system with $\mu = 0$;
$\mathbf{t}(\cdot; \boldsymbol{\Theta}) \leftarrow$ fit a tree to $(\{\mathbf{x}_n\}_{n=1}^N, \mathbf{z}_0)$ (algorithm 2);
**for** $\mu = \mu_0 < \mu_1 < \mu_2 < \cdots < \mu_{\max}$;
  **repeat**
    label-step: $\mathbf{z} \leftarrow$ solve the linear system given fixed $\mathbf{t}(\cdot; \boldsymbol{\Theta})$;
    tree-step: $\mathbf{t}(\cdot; \boldsymbol{\Theta}) \leftarrow$ fit a tree to $(\{\mathbf{x}_n\}_{n=1}^N, \mathbf{z} - \frac{1}{2\mu}\boldsymbol{\lambda})$ given fixed $\mathbf{z}$
      using algorithm 2;
    Lagrange multipliers step: $\boldsymbol{\lambda} \leftarrow \boldsymbol{\lambda} - \mu(\mathbf{z} - \mathbf{t}(\cdot; \boldsymbol{\Theta}))$;
  **until** stop
**end for**
Post-processing (see section 3.3 in the main paper)
**return** $\mathbf{t}(\cdot; \boldsymbol{\Theta})$

---

Figure 1: Pseudocode for LapTAO. "Stop" for inner loop occurs when $(\mathbf{z}, \mathbf{t}(\cdot; \boldsymbol{\Theta}))$ converge (ideally). However, in practice, we use a fixed number of iterations (e.g. 1 in most experiments).

## 2 Derivation of the solution for the label-step

Recall that the *augmented Lagrangian* [8] formulation in eq. (5) in the main paper defines a new, unconstrained optimization problem:

$$\min_{\mathbf{z}} \; \mathcal{L}(\mathbf{z}) = (\mathbf{z} - \mathbf{y})^T \mathbf{J} (\mathbf{z} - \mathbf{y}) + \gamma \, \mathbf{z}^T \mathbf{L} \mathbf{z} - \boldsymbol{\lambda}^T (\mathbf{z} - \mathbf{t}(\mathbf{X}; \boldsymbol{\Theta})) + \mu \|\mathbf{z} - \mathbf{t}(\mathbf{X}; \boldsymbol{\Theta})\|^2. \quad (1)$$

36th Conference on Neural Information Processing Systems (NeurIPS 2022).

```
input training set {(xₙ, yₙ)}ᴺₙ₌₁; penalty α;
    initial tree t(·; Θ) of depth Δ with parameters Θ = {θᵢ}_nodes;
    𝒩₀, …, 𝒩_Δ ← nodes of a tree t(·; Θ) at depth 0, …, Δ, respectively;
generate a reduced set for each node i: ℛᵢ (points that reach node i);
repeat
  for d = Δ down to 0
    for i ∈ 𝒩_d (parallelize, optionally)
      if i is a leaf then
        θᵢ ← take the mean of {yₙ}_{n∈ℛᵢ}
      else
        generate a pseudolabel ȳₙ and weight w̄ₙ = |eₙˡᵉᶠᵗ − eₙʳⁱᵍʰᵗ| for each instance xₙ ∈ ℛᵢ
        θᵢ ← fit ℓ₁ regularized weighted binary classifier on {(xₙ, ȳₙ)} ∈ ℛᵢ with penalty α
    end for
  end for
  update ℛᵢ for each node
until max number of iterations
return t(·; Θ)
```

Figure 2: Tree Alternating Optimization (TAO) algorithm. Here, we limit this pseudocode for decision trees with constant leaves, but it can be trivially extended to any other type. Note: Algorithm 1 for LapTAO uses $\mathbf{z}$ as a ground truth vector (instead of $\mathbf{y}$).

Here, we optimize the problem over $\mathbf{z}$ only and omit the term involving $\alpha$ since we fix the tree $\mathbf{t}(\mathbf{X}; \boldsymbol{\Theta})$ during the label-step. One can see that this is a quadratic function. Next, taking the first and second order derivative w.r.t. $\mathbf{z}$ and multiplying all terms by $0.5$ yields:

$$\frac{\partial \mathcal{L}}{\partial \mathbf{z}} = \mathbf{J}(\mathbf{z} - \mathbf{y}) + \gamma \mathbf{L}\mathbf{z} - \frac{1}{2}\boldsymbol{\lambda} + \mu(\mathbf{z} - \mathbf{t}(\mathbf{X}; \boldsymbol{\Theta})) \quad \text{and} \quad \frac{\partial^2 \mathcal{L}}{\partial \mathbf{z} \partial \mathbf{z}^T} = \mathbf{A} = \mathbf{J} + \mu \mathbf{I} + \gamma \mathbf{L}. \quad (2)$$

It is easy to see that the matrix $\mathbf{A}$ (Hessian) is positive definite, because: $\mu, \gamma > 0$, $\mathbf{L}$ is positive semidefinite [2], $\mathbf{I}$ is identity and $\mathbf{J}$ is the diagonal matrix with first $l$ entries equal 1 and the rest are 0. Therefore, $\mathbf{x}^T \mathbf{A} \mathbf{x} = \mathbf{x}^T \mathbf{J} \mathbf{x} + \mu \mathbf{x}^T \mathbf{I} \, \mathbf{x} + \gamma \mathbf{x}^T \mathbf{L} \mathbf{x} > 0$ for all $\mathbf{x} \in \mathbb{R}^N$. This means that our problem is strictly convex with a unique solution given by the linear system below:

$$\mathbf{A}\mathbf{z} = \mathbf{J}\mathbf{y} + \mu \mathbf{t}(\mathbf{X}; \boldsymbol{\Theta}) + \frac{1}{2}\boldsymbol{\lambda}. \quad (3)$$

Moreover, it is easy to see that $\mathbf{A}$ will be a sparse matrix if graph Laplacian $\mathbf{L}$ is sparse. And this is the case since in practice we construct $\mathbf{W}$ by using the nearest neighbors graph. This allows us to solve the large scale linear system in an efficient way (e.g. via Conjugate Gradient method).

## 3   Accelerating the label-step in LapTAO

Although Conjugate Gradients (CG) method is a reasonable choice to solve the linear system for large scale problems, there is a way to accelerate the label-step for small-medium sized problems. The crucial observation is that the coefficient matrix $\mathbf{A} \in \mathbb{R}^{N \times N}$ is changed by adding $\mu \mathbf{I}$ at each iteration of the Algorithm 1 and the remaining part is static $(\mathbf{J} + \gamma \mathbf{L})$. This naturally leads to the question: can we improve the computation of $\mathbf{A}^{-1}$ from $O(N^3)$ to $O(N^2)$ to solve the linear system (3)? Denote the static part of the matrix as $\mathbf{B} = \mathbf{J} + \gamma \mathbf{L}$. Moreover, $\mathbf{B}$ is a symmetric matrix since $\mathbf{L}$ is symmetric and $\mathbf{J}$ is diagonal. Therefore, we can calculate its eigendecomposition $\mathbf{B} = \mathbf{Q}\boldsymbol{\Lambda}\mathbf{Q}^T$, where $\mathbf{Q}$ is an orthogonal matrix. One can derive the inverse via Sherman-Morrison-Woodbury formula. However, a more direct and easier derivation is:

$$\mathbf{A}^{-1} = (\mu \mathbf{I} + \mathbf{B})^{-1} = (\mu \mathbf{I} + \mathbf{Q}\boldsymbol{\Lambda}\mathbf{Q}^T)^{-1} = (\mathbf{Q}(\mu \mathbf{I} + \boldsymbol{\Lambda})\mathbf{Q}^T)^{-1} = \mathbf{Q}(\mu \mathbf{I} + \boldsymbol{\Lambda})^{-1}\mathbf{Q}^T \quad (4)$$

where $\mu \mathbf{I} + \mathbf{Q}\boldsymbol{\Lambda}\mathbf{Q}^T = \mathbf{Q}(\mu \mathbf{I} + \boldsymbol{\Lambda})\mathbf{Q}^T$ comes from the orthogonality of $\mathbf{Q}$: $\mathbf{Q}\mathbf{Q}^T = \mathbf{I}$. Notice that $\mu \mathbf{I} + \boldsymbol{\Lambda}$ is a diagonal matrix and computing its inverse takes $O(N)$. Therefore, calculating eq. (4) costs $O(N^2)$. The only costly part is computing the eigendecomposition (for $\mathbf{B}$) which still requires $O(N^3)$ time (and destroys the sparsity) but we do it only once before starting our algorithm. In practice, we found this method to be useful only when $N$ is a few thousands at most.

## 4 Experimental setup

### 4.1 Datasets

For all datasets described below, we scale features to have values between 0 and 1, and shift them to be centered around 0. Moreover, we select 1% of training data as cross validation to set the hyperparameters for each method: a tree depth ($\Delta$), confidence threshold for self-training, $\sigma$ and $C$ values for LapSVM, etc. All reported errors are in test sets.

- **mnist** Handwritten digits recognition task [5]. The features are pixel grayscale values in [0,1] of each $28 \times 28$ digit image which belong to one of ten classes. We use the same training/test partition as in [5].

- **susy** Detection of particle collision events (binary classification), available in the UCI Machine Learning Repository [7]. The dataset contains 4.5M points with 18 attributes. We use the first 1M instances and randomly select 90% out of it for training and the rest for test.

- **cpuact** Predict the portion of time that CPUs run in user mode given different system measures. We obtained it from the DELVE data collection[1]. It contains 8192 instances with 21 features. We select 60% of data as training. Since this is a regression task, we provide the output range: $[-0.5, 99.47]$.

- **year_pred** A subset of the Million Song Dataset [3]. The task is to predict the age of a song from several song statistics given as metadata (timbre average, timbre covariance, etc.). The dataset is obtained from the UCI Machine Learning Repository [7]. It has 464k training and 52k test points. The total number of features are 90. Since this is a regression task, we provide the output range: $[1922, 2011]$.

- **fashion_mnist** [9] is another benchmark dataset used for object recognition. It has similar characteristics as mnist (70k grayscale images of $28 \times 28$, 10 classes of different clothing items). We use this dataset primarily to compare LapTAO with LapSVM and to visualize the trained trees. Since the LapSVM has scalability issues for large number of points, we pick the subset of fashion_mnist (3 classes: "shirt", "bag" and "ankle boot") resulting in 18k training points.

### 4.2 Methods

We use TAO to train oblique trees and CART [4] (scikit-learn implementation) to train axis-aligned trees. For all methods that use TAO, we set the total number of TAO iterations to 15. For oblique trees, we tune the following hyperparameters: penalty $\alpha$ and a tree depth $\Delta$. All oblique trees are initialized from a complete tree of depth $\Delta$ with random parameters at each node. As for the axis-aligned trees, these are the hyperparameters to tune: max_depth, min samples at each leaf and min samples to split.

- **LapTAO** we implement our algorithm in Python 3.7.6 and do not use any parallel processing. We tune the following hyperparameters: penalty $\alpha$, tree depth $\Delta$, $k$-nearest neighbors and perplexity parameter $\mathcal{K}$ for the Gaussian affinities. We set the $\gamma = 0.1$ and fix the number of iterations (in Algorithm 1) to 20 starting from small value for $\mu_0 = 0.001$ multiplied by 1.5 after each iteration. The linear system in the label-step is solved either using direct methods (less than 20k dimensions) or Conjugate Gradient method for large scale problems. All trees are initialized from a complete tree of depth $\Delta$ with random parameters at each node.

- **Oblique–all** fits an oblique tree with full supervision (i.e., using all available labeled data), this shows the theoretical maximum performance we can achieve.

- **Oblique–lbl** is the oblique trees trained on labeled portion of data $\mathcal{D}_l$ (this completely discards large portion of unlabeled data).

- Self-training (**axis–self**, **oblique–self**) is an iterative procedure that uses the model predictions to enlarge the portion of labeled data. We closely follow the implementation by

---

[1] `http://www.cs.toronto.edu/~delve/data/comp-activ/desc.html`

Yarowsky [10]. Here, "axis" means traditional axis-aligned trees (trained by CART [4]). We tune the confidence score for classification problems and use all predicted points for regression problems. We set the maximum number of self-training iterations to 10.

- **LapSVM** (Laplacian SVM) is a seminal work by Belkin et al. [2] which has similar problem formulation as LapTAO but for SVM. We tune $\sigma$ and SVM parameter $C$. We use their MATLAB implementation.

- **SSCT** semi-supervised classification trees by Levatić et al. [6]. This method incorporates unlabeled data into greedy tree growing procedure by modifying a splitting criterion to take into account unlabeled data. That is, a splitting score for each [feature, threshold] pair consists of two parts: traditional impurity score (e.g. gini index) for labeled data and a "clustering" score for all available data which pushes each child to have instances of the same cluster. For this method, apart from CART related settings (e.g. max_depth), we tune the weight parameter $w$. We use the R implementation provided in Alabarce et al. [1].

## 5    Additional experimental results

### 5.1    Comparison with SSCT

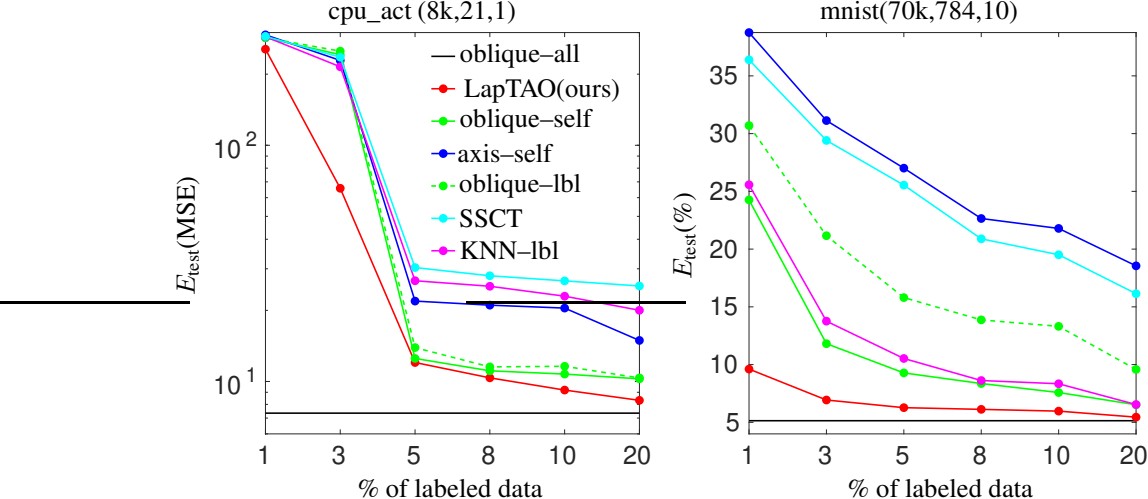

Figure 3: Similar to fig. 2 in the main paper but with SSCT [6] results added (cyan lines). Numbers in brackets report the training size, number of features and output dimension (or number of classes). x–axis shows the percentage of labeled data provided to the algorithm and y–axis shows the test error.

Fig. 3 reports the results of SSCT on the same benchmarks that we used in the main paper. Here, we limit the comparison for cpu_act (regression) and mnist (classification) since other benchmarks took extremely long time (> 7 days) and still failed to complete. Although the original paper [6] considers only classification task, implementation by Alabarce et al. [1] extend it to regression. We did not include this baseline in the main paper to simply avoid cluttering. On top of that, overall results are similar to the "axis–self" and therefore, it does not change our conclusions. Nevertheless, in the left plot (mnist), one can see that SSCT consistently improves over "axis–self" and indeed beneficially leverages information from unlabeled data. However, the performance w.r.t. LapTAO is still noticeably worse. Moreover, "axis–self" shows better error in the regression task (cpu_act).

### 5.2    Comparison with EBBS

We also compare with EBBS (see Table 1), the method which comes from Graph Neural Network (GNN) literature. EBBS is specifically designed for gradient boosted decision trees (GBDT). But by carefully inspecting the work, we do realize that if we limit EBBS to a single tree, then their problem reduces to fitting a tree to the smoothed labels. This is similar to the beginning of the penalty path in LapTAO (initialization step), where we do fit an oblique tree to smoothed labels. However, we

Table 1: Comparing the performance of LapTAO vs EBBS (single tree) on cpu_act (top) and mnist (bottom).

| Method \ % of lbl data | 1% | 3% | 5% | 8% | 10% | 20% |
|---|---|---|---|---|---|---|
| | | | cpu_act | | | |
| LapTAO | 255.13 | 65.75 | 12.03 | 10.36 | 9.19 | 8.32 |
| EBBS (1 tree) | 261.05 | 92.76 | 16.52 | 12.78 | 12.41 | 9.87 |
| | | | mnist | | | |
| LapTAO | 9.61 | 6.93 | 6.27 | 6.12 | 5.97 | 5.45 |
| EBBS (1 tree) | 10.57 | 7.49 | 7.05 | 6.39 | 6.15 | 5.91 |

Table 2: Experimenting with different tree learning algorithms in tree-step of our algorithm: CART vs TAO. For reference, we also report the performances of CART_SELF (CART+self-training) baseline. The results are for cpu_act and the metric used is MSE.

| Method \ % of lbl data | 1% | 3% | 5% | 8% | 10% | 20% |
|---|---|---|---|---|---|---|
| LapTAO | 255.13 | 65.75 | 12.03 | 10.36 | 9.19 | 8.32 |
| LapCART | 263.50 | 83.79 | 18.14 | 13.93 | 13.04 | 11.69 |
| CART_SELF | 293.45 | 228.63 | 21.89 | 21.04 | 20.45 | 14.92 |

would like to point out that the idea of first smoothing the labels throughout the unlabeled points ("label propagation") and then fitting a model ("induction") is well known since the seminal graph-Laplacian SSL approaches, such as references [11–13]. Results in the table above are for cpu_act and mnist, which are clearly better for LapTAO.

## 5.3 Replacing TAO with CART in the tree-step

Table 2 reports the results of using alternative tree fitting algorithms within our framework. Here, LapCART indicates our proposed algorithm but the tree-step was replaced by CART. For reference, we also report the performances of our originally proposed LapTAO and CART_SELF (CART+self-training) baselines. Results clearly indicate superiority of LapTAO over other baselines. They also show that, if we do insist in using CART, our algorithm improves over the CART self-training baseline (CART_SELF). So, it is still beneficial to apply our algorithm with CART.

We would like to emphasize that CART does not support warm-start since it grows a tree from scratch rather than updating the current parameters. This is problematic because CART (and related greedy recursive partitioning algorithms such as C4.5) are known to be very sensitive to the training set: a little change in the data typically leads to completely different tree structures and parameters. Indeed, we observed a significant instability and noisy behavior across iterations with CART. This does not happen with TAO because it takes the tree from the previous iteration as initialization.

## 5.4 Decision tree visualizations

Fig. 4-5 illustrate decision trees obtained from various settings of $\alpha$. The results clearly indicate that this hyperparameter has a direct effect on tree sizes and helps to trade-off between interpretability and model performance. Smaller values for $\alpha$ typically lead to a small error but generates larger trees which may be hard to interpret.