# OpenReview forum: "Semi-Supervised Learning with Decision Trees: Graph Laplacian Tree Alternating Optimization"
_NeurIPS.cc/2022/Conference — NeurIPS 2022 Accept_

### Official Review · Reviewer_sdWW · 2022-06-29

**Rating:** 6
**Confidence:** 5
**Soundness:** 3 good
**Presentation:** 4 excellent
**Contribution:** 3 good

**Summary:**

This work proposes an approach to do semi-supervised learning with decision trees. Their approach splits the objective function into a sparse linear system and a decision tree optimization part. They use alternating optimizations to learn the parameters and minimize the objective function. The linear part is solved analytically. An approximate solution for the tree optimization is obtained using a previously developed TAO method. The authors demonstrate the effectiveness of their method for the SSL tasks on different regression/classification tasks and also analyze the interpretability of their model.

**Questions:**

Questions & Suggestions
1. In the Fig. 1 example, can you also provide any statistical guarantees? For instance, running with different number of parameters, different initializations, tree sizes etc. for LapTAO and other methods. I suspect that with the choice of labels for this 2D data, the classifier can easily choose any of the decision boundaries in the right 2 plots. I will definitely appreciate adding the statistical significance along with the error percentages.
2. Theoretical guarantees: Any ideas of deriving the convergence rate for the alternating optimization? I understand that the tree optimization is NP-Hard but having a probabilistic error term will be very helpful to understand the limitations of this approach.

This is a good direction to explore further. I would like to know the authors views on the above points and their approach to handling them.


**Limitations:**

I request the authors to show results which highlight the limitations of their model. It will be great to get an idea of the size and types of data for which one can opt their method over other SOTA.

**Strengths And Weaknesses:**

Pros:
1. A well thought off application of the Alternating Optimization technique. Since, the TAO algorithm cannot directly optimize the Eq.(1), the reformulation is a neat way of circumventing this problem. [Lines 158-159]
2. The paper is well motivated. The approach is reasonable and the math works out.

Some improvements:
1. Results on larger datasets: Will be interesting to see the results on larger image datasets. I am curious on how the affinity matrix will be designed. Will you take a subset of images which is defined by a batch size? In that case the affinity matrix will need to be recalculated and then the algorithm will be slower. Please share your thoughts and maybe some best practices that you recommend.
2. Comparison with deep learning based approaches: I understand that interpretability is also one aspect to compare. Visualizing the CNN filters for instance in Fig. 4. Also, I would like to see a comparison with SOTA DL methods, that will give the readers an idea about the limitations.

---

> ### Author Response · Authors · 2022-08-01
> **Response to Reviewer sdWW**
>
> > 1. Results on larger datasets...
>
> Please refer to our detailed [response (to all reviewers) above on computational complexity and newly reported runtimes](https://openreview.net/forum?id=cZ41U927n8m&noteId=JWNWEFTYSit). In general, there is no need for mini-batch updates; the graph Laplacian is sparse and computed explicitly, although perhaps using approximate nearest neighbors, in common with standard SSL work using graph Laplacians. See our extended explanation of the computational complexity for further speedups possible. As an example, our algorithm scales well on 1M data points (SUSY) on a regular PC. As for the large scale image datasets, please see our comment on limitations below.
>
> &nbsp;
>
> > 2. Comparison with deep learning based approaches
>
> Thanks for pointing out this possible future research direction. However, this paper is focused on training decision trees with a graph-based SSL objective, which has never been possible before as far as we know. Therefore, (almost) all baselines are tree-based methods.
>
> &nbsp;
>
> > Statistical guarantees...
>
> Fig. 1 is purely for illustrative purposes and to motivate our work. Indeed, for a certain choice of labeled points, the results of naive baseline and LapTAO can be somewhat similar, which is not surprising giving the simplicity of the dataset. The two-moons dataset is another classical example to illustrate SSL. For a certain choice of labeled points, all SSL algorithms will show similar results compared to naive baselines.
>
> &nbsp;
>
> > Theoretical guarantees...
>
> We are not sure what you mean by "probabilistic error term". We can guarantee a monotonic decrease in both steps (label step and tree step) and hence a monotonic decrease of the augmented Lagrangian for fixed multipliers at each iteration. Beyond that, given the NP-hardness of training a tree and the nonconvexity of the overall SSL objective, stronger theoretical guarantees are a difficult but interesting topic of future research.
>
> &nbsp;
>
> > I request the authors to show results which highlight the limitations of their model. It will be great to get an idea of the size and types of data for which one can opt their method over other SOTA.
>
> 1. Please refer to our detailed [response (to all reviewers) above on computational complexity and newly reported runtimes](https://openreview.net/forum?id=cZ41U927n8m&noteId=JWNWEFTYSit). One limitation is that our algorithm is slower than the naive baselines. This is the price to pay for a better model. That said, the algorithm is scalable (linear on the sample size) and can be parallelized in certain parts.
> 2. Tree-based methods are known to work very well with tabular data, but less well than neural nets with image data. Therefore, we did not run our algorithm with large scale image datasets. In such situations, one could still use features from a pretrained deep neural network and then run LapTAO. As of now, we suggest applying LapTAO for mostly tabular data where tree-based frameworks are dominantly used (e.g. XGBoost, Random Forests, etc.).

---

> > ### Comment · Reviewer_sdWW · 2022-08-07
> > **Response to authors**
> >
> > 1. By "probabilistic error term", I meant exploring the Probably Approximately Correct (PAC) learnable conditions, but maybe that's a stretch and can be a potential future direction of exploration.
> >
> > I appreciate the authors detailed responses and applaud them for their hard work. In my opinion, due to the lack of theoretical guarantees, this work needs more rigorous experimental evaluations to substantiate the claims and demonstrate a clear benefit of using LapTAO over the other tree based methods like XGB and others. I will stand by my original evaluation and ratings.

---

### Official Review · Reviewer_dfEo · 2022-07-07

**Rating:** 6
**Confidence:** 4
**Soundness:** 3 good
**Presentation:** 3 good
**Contribution:** 2 fair

**Summary:**

This paper introduces a bi-level optimization method for semi-supervised learning tasks, which iteratively solves a supervised tree learning problem and label smoothing problem. The experimental results demonstrate that LapTAO can achieve accurate and interpretable in label scarcity situations.

**Questions:**

1. Can author provide more details about Graph Laplacian matrix construction? The paper only mentioned Gaussian affinities with k-nearest neighbors, it clusters all available feature vectors?

**Limitations:**

no potential negative societal impact

**Strengths And Weaknesses:**

## Strengths:
The paper is well-written and easy to follow, the reformulation part is well-organized. Some model interpretability are provided, making the model more trustful.


## Weakness:
 1. The framework is not novel to me, [1] also introduces a bi-level optimization method, incorporating label smoothing into decision tree. The author may conduct more literature reviews and comparison with existing methods although [1] and [2] are from Graph Neural Networks (GNNs) community. But most node classification problems in GNNs are semi-supervised learning problem. The paper may have broader impact if the author consider more tasks or datasets from [1] and [2].

2. About model complexity or training time, the author should provide more comparison with other baselines (especially for strong baseline **oblique–self**, the performance is close to LapTAO in Figure 2 ). The author roughly described the complexity and rarely mentioned other baseline's training time.

3. About experimental results, Figure 2 may not be compelling for me, since only LapTAO introduces graph Laplacian matrix and other baselines don't consider graph structure. The gap between LapTAO and other competitors is probably caused by introducing graph information. LapTAO and LapSVM is a fair comparison for me since both methods start from graph Laplacian matrix and the gap between these two methods is narrowed compared with Figure 2. The author may need to carefully conduct a fair baseline comparison or justify why there is a gap between LapTAO with other baselines in Figure 2.

[1]. https://openreview.net/forum?id=nHpzE7DqAnG

[2]. https://openreview.net/pdf?id=ebS5NUfoMKL

---

> ### Author Response · Authors · 2022-08-01
> **Response to Reviewer dfEo (part 1)**
>
> > "1. The framework is not novel to me, [1] also introduces a bi-level optimization method, incorporating label smoothing into decision tree. The author may conduct more literature reviews and comparison with existing methods although [1] and [2] are from Graph Neural Networks (GNNs) community."
>
> Thanks for providing those two references. We were unaware of them, as they are from the Graph Neural Networks literature and upon first inspection appear completely unrelated to our goal (SSL with decision trees). Also, [1] appeared in the beginning of May, which is less than 1 month prior to the NeurIPS deadline (the reviewer guidelines state "Authors are not expected to compare to work that appeared only a month or two before the deadline"). That said, we are happy to comment on this work, as there is indeed a connection. (Work [1] builds on [2] but [2] itself has no relation with our tree SSL focus.)
>
> Firstly, note that "bilevel optimization" refers to an optimization problem having optimization problems in their constraints. This is not our case. Our overall problem (1) is a regular optimization problem but involving a non-differentiable/non-convex function $T$ (the decision tree). We reformulate this as a constrained problem in (2)-(3), but this is still a regular, not bilevel optimization. Finally, we apply the augmented Lagrangian and alternating optimization to the constrained problem, resulting in our actual algorithm.
>
> Now regarding your reference [1]. As mentioned, this is a GNNs work which involves a graph prior in the objective and an algorithm (EBBS) specifically designed for gradient boosted decision trees (GBDT). By carefully inspecting [1], we do realize that if we limit EBBS to a single tree, then their problem reduces to fitting a tree to the smoothed labels. This is similar to the beginning of the penalty path in our algorithm (our initialization step), where we do fit our oblique tree (using TAO) to smoothed labels. Please refer to section 3 and pseudocode 1 in the suppl. mat. Note however that the basic idea of first smoothing the labels throughout the unlabeled points ("label propagation") and then fitting a mapping to them ("induction") is well known since the seminal graph-Laplacian SSL approaches, such as references [24-26] in our paper.
>
> Importantly, our algorithm then alternates between label smoothing and tree fitting, while increasing the penalty parameter $\mu$, as a way to optimize our overall problem (1) jointly over the labels and the tree. Let us see this works better. Although [1] did not provide an implementation, we did our best to implement the version of EEBS for a single tree using their provided algorithm and experimental setup. Here are our results for cpu_act and MNIST, which are clearly better for LapTAO:
>
> **CPU_ACT:**
> |Method \ % of lbl data |   1%   |   3%   |   5%  |   8%  |  10%  |  20%  |
> |----------------------------|---------|----------|---------|--------|---------|----------|
> |LapTAO               | 255.13 | 65.75  | 12.03 | 10.36 | 9.19  | 8.32  |
> |EBBS (1 tree)        | 261.05 | 92.76  | 16.52 | 12.78 | 12.41 | 9.87  |
>
> &nbsp;
>
> **MNIST:**
> |Method \ % of lbl data |  1%   |  3%  |  5%  |  8%  | 10%  | 20%  |
> |----------------------------|---------|--------|--------|-------|--------|---------|
> |LapTAO               | 9.61  | 6.93 | 6.27 | 6.12 | 5.97 | 5.45 |
> |EBBS (1 tree)        | 10.57 | 7.49 | 7.05 | 6.39 | 6.15 | 5.91 |
>
> &nbsp;
>
> About running LapTAO datasets from [1,2]. We agree that those are interesting potential applications and exploring LapTAO could be worth trying. However, in terms of dataset characteristics (size, feature dimensions, number of edges), we believe our benchmarks are on the same level and even much bigger; and we also cover both classification and regression problems. For instance, the largest dataset in [1,2] has 54k instances (and most of them use small number of features), whereas we ran our algorithm on datasets up to 1M points.
>
> &nbsp;
>
> > 2. Regarding training runtime
>
> Please see our detailed [response (to all reviewers) above on computational complexity, scalability and reported runtimes](https://openreview.net/forum?id=cZ41U927n8m&noteId=JWNWEFTYSit).

---

> > ### Author Response · Authors · 2022-08-01
> > **Response to Reviewer dfEo (part 2)**
> >
> > > 3. About experimental results... other baselines don't consider graph structure
> >
> > - We did include two baselines that consider a graph structure: LapSVM and SSCT (see suppl. mat. Figure 3). SSCT uses a "clustering" score at each tree split which is based on a neighborhood graph. We will move the comparison with SSCT to the main paper if there is space. Regarding LapSVM, the performance is comparable or better (especially when \% of labeled data is small). This is remarkable given that we achieve this using a single oblique tree whereas LapSVM uses a kernel SVM. Finally, see our additional comparison with EBBS (1 tree) above.
> >
> > - Also, as a generic comment, incorporating graph structure in tree learning is not easy task and the remaining baselines do not use this information. Otherwise, this problem would have been solved a long time ago. Therefore, other baselines had to use various heuristics (e.g. self-training) to enable SSL for trees. In this paper, we believe we are the first to show that it is possible to define a training objective function for a decision tree that incorporates a graph prior and optimize it (approximately). Since our focus is on learning a single tree within the SSL framework, we've included all tree-based SSL baselines.
> >
> > &nbsp;
> >
> > > Can author provide more details about Graph Laplacian matrix construction? The paper only mentioned Gaussian affinities with k-nearest neighbors, it clusters all available feature vectors?
> >
> > We use the same approach as for constructing affinities for t-SNE with a fixed perplexity $K$ (Hinton and Roweis, Stochastic Neighbor Embedding, NIPS 2003). Basically, it is similar to regular Gaussian affinities but the bandwidth $\sigma$ is set individually for each training point such that it has a distribution over neighbors with perplexity $K$. We describe it further in the suppl. mat. (section 3.2). Such affinities are widely used in non-linear dimensionality reduction and we find it works better in our experiments. More generally, the graph construction really depends on the problem and practitioners may want to explore other possibilities (e.g. string kernels might work better for NLP related problems).

---

> > > ### Comment · Reviewer_dfEo · 2022-08-09
> > > **Thanks for your response.**
> > >
> > > I appreciated the author detailed response. Most questions have been resolved, and author should add EBBS for comparison in the revised version. I raised my score to 6.

---

### Official Review · Reviewer_bXo9 · 2022-07-09

**Rating:** 6
**Confidence:** 4
**Soundness:** 3 good
**Presentation:** 3 good
**Contribution:** 2 fair

**Summary:**

This paper studies the semi-supervised decision tree learning problem. By introducing additional variables $z$, the original problem can be reformulated and solved iteratively as two simpler problems: a label smoothing problem, which can be solved through a sparse linear system, and a supervised tree learning problem, which was solved by the Tree Alternating Optimization algorithm in this paper. The numerical experiments demonstrate that the proposed method in this paper is able to achieve high testing accuracy and good interpretability.

**Questions:**

1. Since TAO method in "tree-step" requires solving the logistic regression on the whole training set over all branch nodes for multiple iterations, for large dataset it can be very time consuming. For that reason, I do not expect the LapTAO algorithm can be run for too many iterations within a particular time limit. My question is, have the authors tried to use other more efficient algorithms (can be greedy) to fit the tree?
2. Computation time should also be reported in the experiment section.

**Limitations:**

1. The main LapTAO algorithm has no theoretical analysis.
2. The selection of benchmark methods in the experiment part is not representative and seems very random. For example, the authors do not have to compare with KNN or any other axis-parallel decision tree, instead, the authors should focus on comparing the performance of LapTAO with the application of other semi-supervised learning algorithm on decision tree (e.g., SSCT in the supplementary materials). Similarly, the authors can implement experiments on the extension of LapTAO to other models, and compare with the successful semi-supervised learning algorithms of those models.

**Strengths And Weaknesses:**

Strength:
1. This paper studies semi-supervised version of decision tree learning problem, which has not been well-studied so far.
2. The algorithm LapTAO introduced in this paper can be easily extended to classification and multioutput regression problems, and can even be extended to other models with little modification.

Weakness:
1. The main algorithm LapTAO is essentially applying ADMM with coordinate descent to the original problem (1).
2. The choice of using TAO method for "Tree-step" does not quite make sense to me, this seems arbitrary and the reason is not fully justified. There are many other efficient algorithms for training optimal decision tree, with or without regularization term.

---

> ### Author Response · Authors · 2022-08-01
> **Response to Reviewer bXo9 (part 1)**
>
> > The main algorithm LapTAO is essentially applying ADMM with coordinate descent to the original problem (1).
>
> Thanks for pointing out. However, we are not sure why reviewer listed this as a weakness: we solve a long-standing problem (graph SSL with trees) in a convenient way using modern optimization techniques. Also, our proposed approach is not quite ADMM. The problem in (1) is not directly in a form where we can apply ADMM. We believe that reformulating the problem by introducing new variables and consequently reducing it to a linear system and tree fitting is non-trivial. Moreover, there are other important contributions in our method: exact solution for the leaves given the fixed decision nodes (section 3.3), deriving the solution for the beginning of the $\mu$ path (i.e., initialization), accelerating the "label step" (section 1.2 in the suppl. mat.), etc.
>
> &nbsp;
>
> > The choice of using TAO method is not justified...
>
> It is true, and an advantage of our approach, that (in principle) we can use any other tree learning algorithm in the tree step. But there are important reasons for using TAO:
> 1. TAO supports axis-aligned and oblique trees.
> 2. TAO is able to find much better optima of the tree objective function in a scalable way. Critical for this is the fact that TAO guarantees a monotonic decrease of the objective at each iteration (see reference [4] in the main paper).
> 3. TAO can handle sparsity regularization terms for oblique trees, such as an L1 penalty. This generates highly accurate but relatively shallow trees that use few nonzero weights, which leads to high interpretability.
> 4. TAO supports warm start, i.e., the ability to improve over a given tree (from the previous iteration), which is important to save runtime (by providing a good initialization) and also to avoid oscillations (switching erratically to different local optima). The ability to use warm-start is a rather unique property among tree-learning algorithms. Indeed, the traditional, greedy recursive partitioning algorithms such as CART or C4.5 do not support warm start; instead, they induce a new tree from scratch at each iteration. This is problematic because it leads to significant instability and noisy behavior across iterations. (It is well known that a little change in the training data often leads to completely different tree structures and parameters in these algorithms; see Peter Turney: "Technical Note: Bias and the Quantification of Stability". Machine Learning(20):23-33, 1995.) Another type of tree-learning algorithms are based on mixed integer optimization (brute-force search, typically via a branch-and-bound algorithm). These algorithms do not support warm-start either, but more importantly, their worst-case runtime is exponential and they do not scale beyond tiny trees (depth up to 4) and tiny datasets -- the latter making them wholly unsuitable for SSL, obviously. They also have significant restrictions, usually requiring binary (not continuous) features and axis-aligned trees only.
> 5. And last but not the least: it works well in practice, as we convincingly show in the experiments section.
>
> To confirm the importance of using TAO with sparse oblique trees, we experimented with other algorithms in the "tree--step" of our algorithm. For instance, consider below results for CPU_ACT:
>
> |Method \ % of lbl data |   1%   |   3%   |   5%  |   8%  |  10%  |  20%  |
> |------------------------------|----------|---------|--------|---------|-------|-----------|
> |LapTAO               | 255.13 | 65.75  | 12.03 | 10.36 | 9.19  | 8.32  |
> |LapCART              | 263.50 | 83.79  | 18.14 | 13.93 | 13.04 | 11.69 |
> |CART_SELF            | 293.45 | 228.63 | 21.89 | 21.04 | 20.45 | 14.92 |
>
> Here, LapCART indicate our proposed algorithm but "tree$-$step" was replaced by CART. For reference, we also report the performances of our originally proposed LapTAO and CART_SELF baselines. Results clearly indicate superiority of LapTAO over other baselines. They also show that, if we do insist in using CART, our algorithm improves over the CART self-training baseline.
>
> &nbsp;
>
> > The main LapTAO algorithm has no theoretical analysis.
>
> The reviewer should realize of the difficulty of the computational problem: training a decision tree on its own is NP-hard, and the graph-Laplacian SSL learning problem is heavily nonconvex and nondifferentiable due to the model being a tree. That said, we can guarantee a monotonic decrease in both steps (label step and tree step) and hence a monotonic decrease of the augmented Lagrangian for fixed multipliers at each iteration. Given the competitive empirical performance of our algorithm, stronger theoretical guarantees are an interesting topic of future research.

---

> > ### Author Response · Authors · 2022-08-01
> > **Response to Reviewer bXo9 (part 2)**
> >
> >
> > > Scalability, computational complexity and exact runtimes...
> >
> > Please see our detailed [response (to all reviewers) above on computational complexity and reported runtimes](https://openreview.net/forum?id=cZ41U927n8m&noteId=JWNWEFTYSit). Importantly, as noted in our explanation there, note that the "tree$-$step" does **not** require solving the logistic regression on the whole training set over all branch nodes in each TAO iteration. We only run $\Delta$ (tree depth) logistic regressions on the whole training set in total in each TAO iteration.
> >
> > &nbsp;
> >
> > > "The selection of methods in the experiment part is not representative and seems very random..."
> >
> > We've included all tree-based SSL baselines (SSCT, self-training, fully supervised tree) that we are aware of. Furthermore, we've extended self-training for oblique trees and for the regression setting to make an apples-to-apples comparison. Unfortunately, SSCT does not support oblique trees and extending it requires significant modification to the implementation. Moreover, SSCT has scalability issues even with axis-aligned tree which will get even worse with oblique splits. Even more, we went further and compared against other methods (non tree-based) that also apply manifold regularization (e.g. LapSVM). The performance is comparable or better (especially when the proportion of labeled data is small). This is remarkable given that we achieve this using a single tree, whereas LapSVM uses a kernel SVM. We explain the motivation and reasons behind choice of baselines in section 4.1.
> >
> > &nbsp;
> >
> > > "Similarly, the authors can implement experiments on the extension of LapTAO to other models..."
> >
> > Extending LapTAO to other models (e.g. neural nets, GBDT) is straightforward but beyond the scope of this paper. Our focus was on making the optimization of a graph-based SSL objective possible, for the first time, with decision trees.

---

> > > ### Comment · Reviewer_bXo9 · 2022-08-09
> > > **Update of the score**
> > >
> > > I appreciate the authors' detailed explanations, and my main concern has been addressed. For that reason I decide to raise my score to 6.

---

### Official Review · Reviewer_e5V7 · 2022-07-11

**Rating:** 8
**Confidence:** 5
**Soundness:** 4 excellent
**Presentation:** 4 excellent
**Contribution:** 4 excellent

**Summary:**

Many real-world machine learning applications encounter both labeled and unlabeled data.
In such problems there may be a scarcity of high-quality labeled data, while large amounts of unlabeled or partially labeled data are available.
Semi-supervised learning (SSL) involves the task of training machine learning models on such datasets to obtain more accurate predictions than models trained on labeled data alone.
Graph-based methods for regularization have been demonstrated to provide efficient methods for  SSL problems, including the popular Laplacian regularized least squares and Laplacian support vector machines (LapSVM).
However, extending graph-based regularization to the problem of training trees in SSL tasks has received comparatively less attention.

This paper proposes an algorithm for training decision trees in SSL problems. The proposed approach combines ideas from manifold regularization, sparse regularization, and tree optimization to obtain a sample efficient algorithm for training sparse decision trees.
The algorithm alternates between a "label-step'' and a "tree-step'' until a stopping criterion such as maximum number of iterations or convergence is satisfied.
In the label-step the augmented Lagrangian method is applied, yielding an unconstrained optimization problem whose solution yields an updated approximation of the labels based on the current (fixed) tree.
In the tree-step, tree alternating optimization (TAO) is applied to update the parameters of the decision tree to fit the current estimates of the labels.

The authors apply the resulting LapTAO (for Laplacian TAO) algorithm to the training of oblique decision trees in a variety of tasks. The results show that oblique trees trained with LapTAO obtain smaller MSE on the test set and testing error compared to a variety of alternative approaches for training axis-aligned and oblique trees in SSL, as well as KNN on the labeled data alone. The authors also demonstrate LapTAO yields similar performance to LapSVM on the Fashion-MNIST problem given greater than 3% of training data, while outperforming that method by considerable margin when trained with less than 3% labeled data. Finally, the authors display a visualization of the sparse parameter vectors obtained by the algorithm in the Fashion-MNIST case to demonstrate the interpretability of the resulting oblique decision tree.


**Questions:**

Line 330: in the description of Figure 4 it says "$\alpha = 1$ and $\alpha=0.1$'', should the second value be $\alpha=10$? Same also on line 331, see the caption of Figure 4.

Supplemental material, line 13: Should it say "$\mathbf{L}$ is positive" instead of "$\mathbf{J}$ is positive''?

Why was only 1 iteration used in most of the experiments, as described in the caption of the pseudocode in Figure 1 of the supplemental material? Were other values tried?



**Limitations:**

The authors addressed the limitations and potential negative societal impact of the work.

**Strengths And Weaknesses:**

The paper is well-written, and the proposed approach is interesting and novel. The derivations are simple and straightforward, and the presentation and length of the paper are good. In particular, Section 3 of the paper is easy to follow and the authors provide ample discussion on initialization, extension of LapTAO to classification and multioutput regression problems, and extension of the general proposed alternating approach to other models such as neural networks. The experiments also provide many comparisons to other approaches, demonstrating the LapTAO algorithm achieves higher accuracy over other approaches for SSL, with a focus on tree-based methods in particular but also including LapSVM, on a variety of tasks. SSL is an important problem relevant to many application areas and decision trees are also an important direction of active research.

However, the paper lacks somewhat in the depth of the experimental validation of the claims of the improved performance over other approaches for SSL and in discussion of computational complexity of the approach in large scale machine learning tasks. The experiment results presented are for the MNIST, susy, cpuact, year\_pred, and fashion-MNIST datasets. As the authors describe, the largest experiment was conducted on a CPU and took less than 24 hours. It would be interesting to see how the LapTAO method performs on some larger scale benchmark prediction tasks where GPUs are necessary, in particular with regard to the computational complexity relative to other popular approaches. It would also be interesting to see further comparisons on the proposed regularization approach with different types of trees, for example using this formulation with axis-aligned trees and CART.

Section 4.3 on model interpretability displays visualizations of the weights of the nodes of some oblique trees trained with LapTAO on the Fashion-MNIST dataset. While sometimes it appears clear from the displayed weights why the model takes one branch over another (e.g. in the case of the boot from node 1 of the $\alpha=10$ case of Figure 4), other times it is not so clear. It would be interesting to see a comparison with models trained without the $\ell_1$ term, i.e. $\alpha = 0$. Overall, the discussion on model interpretability and sparsity could be improved.

---

> ### Author Response · Authors · 2022-08-01
> **Response to Reviewer e5V7**
>
> > Line 330: in the description of Figure 4 it says "$\alpha=1$ and $\alpha=0.1$", should the second value be $\alpha=10$? Same also on line 331, see the caption of Figure 4.
>
> > Supplemental material, line 13: Should it say "$\textbf{L}$ is positive" instead of "$\textbf{J}$ is positive"?
>
> Thanks for noticing these typos! Yes, the second value should be $\alpha=10$ and the first matrix in suppl. mat. should be $\textbf{L}$. We will correct that.
>
> &nbsp;
>
> > Why was only 1 iteration used in most of the experiments, as described in the caption of the pseudocode in Figure 1 of the supplemental material? Were other values tried?
>
> For each fixed value of $\mu$, (ideally) we should iterate (tree$-$step and label$-$step) until convergence and we investigated this for toy 2D dataset and on MNIST. Specifically, we tried >10 iterations per $\mu$ and it only marginally improved the overall performance but significantly increased the runtime. Therefore, we kept only one iteration for the remaining set of experiments which is commonly practiced in other applications of augmented Lagrangian.
>
> &nbsp;
>
> > Regarding tree visualizations and interpretability: a tree with $\alpha=0$ and "the discussion on model interpretability and sparsity could be improved..."
>
> Thanks! We will add more discussion (together with more elaborate analysis of our obtained tree) in the next revision. Our main point in that section was to show that we can inspect and verify our final model trained using SSL (interpretable semi-supervised learning?). We can ask such questions as: does the unlabeled data was able to improve our model? Whether it leads to a better feature utilization? Are the features used by the tree meaningful? etc. We believe this is extremely important for sensitive applications such as credit scoring where regulations are strict (GDPR, AI act, etc.)
>
> &nbsp;
>
> > Regarding scalability and GPU
>
> Please see our detailed [response (to all reviewers) above on computational complexity and reported runtimes](https://openreview.net/forum?id=cZ41U927n8m&noteId=JWNWEFTYSit). Regarding GPU implementation, we are thinking about this avenue as future research direction. Indeed, leveraging GPU can significantly boost the runtime but, unfortunately, implementing trees in GPU is non-trivial due to bad memory locality. That said, there are some research done in this direction (e.g. GPU version of XGBoost). In our current implementation, the simplest thing to consider would be GPU acceleration of the logistic regression training for large scale data.
>
> &nbsp;
>
> > In "tree--step", replace TAO oblique with CART axis-aligned
>
> We indeed experimented with other algorithms in the "tree$-$step" of our algorithm. For instance, consider below results for CPU_ACT:
>
> |Method \ % of lbl data |   1%   |   3%   |   5%  |   8%  |  10%  |  20%  |
> |----------------------------|---------|----------|---------|--------|---------|----------|
> |LapTAO               | 255.13 | 65.75  | 12.03 | 10.36 | 9.19  | 8.32  |
> |LapCART              | 263.50 | 83.79  | 18.14 | 13.93 | 13.04 | 11.69 |
> |CART_SELF            | 293.45 | 228.63 | 21.89 | 21.04 | 20.45 | 14.92 |
>
> Here, LapCART indicate our proposed algorithm but "tree--step" was replaced by CART. For reference, we also report the performances of our originally proposed LapTAO and CART_SELF baselines. Results clearly indicate superiority of LapTAO over other baselines. That said, it is still beneficial to apply our algorithm with CART since it improves over self-training baseline.
>
> Another important thing is that CART does not support warm-start at each iteration, because it grows a tree from scratch from the root rather than update the current tree's parameters. This is problematic because CART (and related greedy recursive partitioning algorithms such as C4.5) are known to be very sensitive to the training set: a little change in the data typically leads to completely different tree structures and parameters. Indeed, we observed a significant instability and noisy behavior across iterations with CART. This does not happen with TAO because it takes the previous iteration's tree as initialization.

---

> > ### Comment · Reviewer_e5V7 · 2022-08-08
> > **Update 2022-08-08**
> >
> > I agree generally with author remarks about comparisons with deep learning approaches. The point of the paper is on training decision trees with graph-based SSL objective. Future works should explore these comparisons, but I agree that this is beyond the scope of the current study.
> >
> > Authors should add the discussion on GPU implementations raised in our comments. Some implementations may yield improvements to the overall runtimes. Also, comments from rebuttal about parallelization are interesting and could be added to revised submission. These combined suggest fruitful directions of future research for proposed approach. I also agree with the authors that scaling with respect to sample size is an appropriate metric to guage performance in large-scale SSL problems. Comments about limitations of trees on non-tabular data should be added for more complete picture of limitations of the proposed approach. I agree with the authors about the complexity discussion, and that this should be expanded in the texts as per reviewer recommendations. Also recommend moving comparison to SSCT from supplement to main paper (space permitting).
> >
> > I agree with Reviewer sdWW that the discussion about theoretical guarantees could be slightly. Stronger guarantees are likely difficult to obtain, as the authors point out, and the derivations and experimental results presented are a good first step. I recommend expanding this discussion per sdWW’s comments and with author response.
> >
> > Per reviewer dfEo’s recommendations, some of the comparisons from [1] and [2] could have been added, although I’m not sure that they would reveal much. Agree with authors about the GNN references comparison not being directly applicable. Reference [1] appeared within the “not required to compare” timeframe. Also agree with rebuttal about bi-level optimization. Overall, these references are closely related, but do not cover the same topic. Authors should include the comparisons to [1] and [2] from their rebuttal in the revision.
> >
> > After reading the other reviewers' comments and author responses, I update my score to 6-7.

---

### Author Response · Authors · 2022-08-01
**Concerns raised by several reviewers**

We thank all reviewers for their valuable comments and for the effort they have put in our paper. Below we provide responses to the generic comments raised by several reviewers:

> **Computational complexity and scalability**

Although the paper (page 5) had a paragraph about the computational complexity, we realize it was too terse and we'll expand it. The most important thing to notice is that the complexity is linear on the sample size $N$. In detail, at the top level, LapTAO runs a problem-dependent number of iterations (depending on the $\mu$ schedule, typically less than 20). Each iteration has to solve two subproblems (approximately):

- The label step: this is a large, sparse linear system of $N \times N$, solved approximately with Conjugate Gradients, initialized by the previous iterate (warm-start). Each CG iteration is $O(N k)$ where $k$ is the average number of neighbors in the graph, and the number of CG iterations is at most $N$ (in practice, convergence occurs much faster). The total runtime of the label step is less than 30 seconds in the largest experiment we conducted (1GB of data, 1M points). Convergence can be further improved via preconditioning (e.g. Jacobi). (We can also solve the linear system exactly in $O(N^2)$ by caching its SVD, as noted in the supplementary material section 1, but this is only convenient if $N$ is a few thousands at most.)

- The tree step: fitting an oblique tree with TAO to the $N$ training points. Each iteration of TAO updates each decision node and leaf node. For each leaf, we compute the average of its reduced set (training points reaching it), so this is $O(N D)$ over all the leaves (since their reduced sets total $N$ points and $D$ is the feature dimension). For each decision node, we train a logistic regression on its reduced set. Assuming logistic regression is linear on the sample size and dimensionality, this is also $O(N D)$ for all decision nodes at the same depth (since, as for the leaves, their reduced sets total $N$ points), although with a larger constant factor in the big-O notation than for the leaves. Hence, processing all the decision nodes in the tree is $O(\Delta N D)$, or equivalently, running $\Delta$ logistic regressions on the whole training set. See more details in [4]. Importantly (regarding a question by **reviewer bXo9**), note that the cost is not that of solving a logistic regression on the whole training set *over all decision nodes* (of which there are $2^{\Delta}-1$); we only run $\Delta$ logistic regressions. This is a critical advantage of TAO and is due to the fact that each node (decision or leaf node) only handles the points in its reduced set. In summary, the overall runtime of TAO is $O(\Delta D N)$ per TAO iteration (we run 10 TAO iterations in our experiments).

Since the tree step dominates the label step, in terms of runtime, our algorithm is almost like sequentially training decision trees (as in boosting), which is what the self-training baseline does.

There are two additional speedups which we did not explore here. First, TAO itself can be parallelized depth-wise, i.e., the nodes at the same depth (whose reduced sets are disjoint) can be optimized in parallel. Our implementation and runtimes in the paper are purely sequential. Second (regarding a question by **reviewer e5V7**), using GPUs. This is possible with GPU-friendly implementations of logistic regression, and also because oblique trees involve scalar products (unlike axis-aligned trees). In general, GPU acceleration of oblique tree algorithms is an interesting topic worthy of research.

Finally, this computational cost should also include computing the nearest-neighbour graph (and the graph Laplacian L). This is indeed a large cost. A naive implementation requires $O(D*N^2)$ to calculate the distance vector for each point and determine the nearest neighbors. This can be parallelized for each point. With large datasets, one can perform approximate nearest neighbors search (e.g. via Locally Sensitive Hashing). However, the graph construction is orthogonal to our work, and leveraging the graph Laplacian is by far the standard approach in the SSL literature.

&nbsp;

> **Exact training runtimes on benchmark datasets (section 4.2)**

|Dataset\Method | LapTAO (ours) | oblique-self | axis-self | SSCT   |
|----------------------|-------------|-----------------|----------|------------|
|cput_act       | 1072s  |    934s      |   23s     | 936s   |
|mnist          | 11027s |    9572s     |   514s    | 15932s |
|SUSY           | 22421s |    17873s    |   816s    | >1d    |

**Note**: we ran our code on a regular PC, with little parallel processing and using unoptimized Python implementation. Therefore, the training runtime for LapTAO can be significantly improved.

---

### Meta-Review · Area_Chair_uTsb · 2022-08-27

**Recommendation:** Accept
**Confidence:** Certain

**Metareview:**

This paper extends graph-based semi-supervised learning to decision tree classifiers, where the optimization gets much more challenging.  The proposed solution reformulates the problem with a new auxiliary variable, which leads naturally to an iterative solution of alternating between 1) supervised learning on trees, and 2) label smoothing via a sparse linear systems.  High accuracy is favorable interpretability of the method are demonstrated in numerical experiments.

All the reviewers, including myself, find the paper a solid contribution to the methodology and analysis. There are a few concerns such as computational complexity, and the rebuttal has done a good job addressing it (and other concerns).  These additional results and insights can be included in the final version of the paper.

**Award:**

No

---

### Decision · Program_Chairs · 2022-09-14

Accept